# Structure-guided disulfide engineering restricts antibody conformation to elicit TNFR agonism

Isabel G. Elliott [1,2,3,4,7], Hayden Fisher [1,2,3,4,5,7], H. T. Claude Chan[3], Tatyana Inzhelevskaya[3], C. Ian Mockridge[3], Christine A. Penfold[3], Patrick J. Duriez[3], Christian M. Orr [6], Julie Herniman [1], Kri T. J. Müller [3], Jonathan W. Essex [1,4,8], Mark S. Cragg[3,4,8] & Ivo Tews [2,4,8] ✉

A promising strategy in cancer immunotherapy is activation of immune signalling pathways through antibodies that target co-stimulatory receptors. *h*IgG2, one of four human antibody isotypes, is known to deliver strong agonistic activity, and modification of *h*IgG2 hinge disulfides can influence immune-stimulating activity. This was shown for antibodies directed against the *h*CD40 receptor, where cysteine-to-serine exchange mutations caused changes in antibody conformational flexibility. Here we demonstrate that the principles of increasing agonism by restricting antibody conformation through disulfide modification can be translated to the co-stimulatory receptor *h*4-1BB, another member of the tumour necrosis factor receptor superfamily. Furthermore, we explore structure-guided design of the anti-*h*CD40 antibody ChiLob7/4 and show that engineering additional disulfides between opposing F(ab') arms can elicit conformational restriction, concomitant with enhanced agonism. These results support a mode where subtle increases in rigidity can deliver significant improvements in immunostimulatory activity, thus providing a strategy for the rational design of more powerful antibody therapeutics.

Canonical antibodies comprise two identical polypeptide light chains and two identical polypeptide heavy chains. The Fc domain provides long half-life and effector function, whereas the antigen-binding F(ab') regions imbue antibodies with exquisite specificity, with these two determinants linked by the hinge region[1–3]. These inherent properties of antibodies have been exploited extensively for laboratory assays, diagnostics, and therapeutics. Their success in the latter is judged by the fact that there are currently over 200 antibody therapeutics

approved or in regulatory review, with >90 indicated for cancer treatment[4,5].

Therapeutic antibodies can be classified according to their principal mechanisms of action[6]. One class of reagents are immunomodulatory antibodies, which function by targeting receptors on immune cells. Within this group, antibodies targeting immune checkpoint receptors, such as PD-1, PD-L1 and CTLA-4, (so-called checkpoint blockers) have delivered transformative success in the clinic as cancer

[1]School of Chemistry and Chemical Engineering, Faculty of Engineering and Physical Sciences, University of Southampton, Southampton SO17 1BJ, UK. [2]School of Biological Sciences, Faculty of Environmental and Life Sciences, University of Southampton, Southampton SO17 1BJ, UK. [3]Antibody & Vaccine Group, Centre for Cancer Immunology, School of Cancer Sciences, Faculty of Medicine, Southampton General Hospital, University of Southampton, Southampton SO16 6YD, UK. [4]Institute for Life Sciences, University of Southampton, Southampton SO17 1BJ, UK. [5]European Synchrotron Radiation Facility, Grenoble, Cedex 9 38043, France. [6]Diamond Light Source, Didcot OX11 0FA, UK. [7]These authors contributed equally: Isabel G. Elliott, Hayden Fisher. [8]These authors jointly supervised this work: Jonathan W. Essex, Mark S. Cragg, Ivo Tews. ✉e-mail: Ivo.Tews@soton.ac.uk

therapeutics in previously difficult to treat cancers[7,8]. A second group of antibodies, known as immunostimulatory antibodies, have been proposed as alternative and complementary anti-cancer therapeutics[9]. These reagents, often targeting co-stimulatory molecules, such as 4-1BB, CD40, OX40, and CD27, which belong to the tumor necrosis factor receptor superfamily (TNFRSF), have demonstrated impressive pre-clinical activity and proof of concept[9–11]. Through targeting and binding co-stimulatory molecules expressed on immune cells, immunostimulatory antibodies are capable of eliciting receptor clustering and subsequent downstream signaling of intracellular immune pathways[11,12]. This can evoke cellular activation to provide powerful anti-tumor responses[13]. Several of these antibodies have reached clinical trials, including two anti-4-1BB mAb, Utomilumab and Urelumab[10]. However, these mAb eventually failed in the clinic due to problems with efficacy and toxicity, respectively[14].

The rational design of agonistic antibodies has been explored to enhance the biological activity of these reagents. Approaches such as modulating affinity[15], increasing valency[16], optimizing Fc gamma receptor (FcγR) interactions[17], switching isotype[18,19], and modifying hinge structure[20] have all been utilized to manipulate the agonistic activity of immunostimulatory antibodies[10]. The hinge is of particular interest as many of the differences in the four isotypes of human (h) IgG lie in this region. The hinge length, amino acid composition, and disulfide bond pattern all vary between hIgG isotypes, and the resulting differences in hinge structure are responsible for the wide range of effector activity observed across the isotypes[21]. However, less well appreciated was the impact of the hinge on antibody-mediated receptor agonism.

We previously showed that of the four human isotypes, hIgG2 provides the strongest agonism and greatest receptor clustering for monoclonal antibodies (mAb) targeting several members of the TNFRSF (including 4-1BB, CD40, OX40, and CD27) as well as the immunoglobulin superfamily member CD28[12,19,22,23]. The hIgG2 isotype even demonstrated the ability to convert clinically relevant anti-hCD40 hIgG1 antagonists into powerful agonists[18]. For this, hinge swap experiments, placing the hIgG2 hinge into hIgG1 and vice versa, established the hIgG2 hinge as the primary determinant of activity.

The hIgG2 hinge is unique in that it can naturally undergo disulfide shuffling in the blood via a redox mechanism, with the IgG2-A and IgG2-B isoforms representing the extremes in hinge disulfide structure[24–27]. In several anti-hCD40 mAb, these two isoforms have opposing levels of immunostimulatory activity, with hIgG2-A inactive and hIgG2-B strongly agonistic[19]. We have probed the effects of hinge disulfide variation on agonistic activity using site-directed mutagenesis of the hIgG2 hinge region of the clinically relevant anti-hCD40 mAb ChiLob7/4[28]. Cysteine-to-serine (C-S) variants were ordered along a hierarchy of biological activity and then characterized structurally. For the most biologically active variant, C232S κC214S, disulfide linkages between C127 and C233 on opposing heavy chains were identified and termed a disulfide cross-over (amino acid numbering follows Kabat nomenclature, κ stands for the κ-light chain, see Methods). Using in-solution small angle X-ray scattering (SAXS) analysis combined with molecular dynamics (MD) simulations, we demonstrated that the cross-over led to conformational restriction, while variants with fewer hinge disulfides and lacking this cross-over were highly flexible, such as the inactive hIgG2 C232S + C233S variant (Fig. 1a)[20].

Here, we translate the concept of activity modulation through hIgG2 hinge engineering to the co-stimulatory immune receptor 4-1BB. h4-1BB is able to provide proliferative, survival, and cytotoxic signals to T cells, and thus has been co-opted into chimeric antigen receptor designs[14], as well as targeted with various antibody modalities, some of which have progressed to clinic[10]. Therefore, being able to modulate the activity of this receptor through antibody hinge engineering is an attractive proposition. We show that hinge region C-S exchange

mutations affect antibodies directed against hCD40 and h4-1BB in a similar manner, where increased antibody compactness is associated with higher agonistic activity. We then demonstrate that it is possible to generate antibody variants against hCD40 with additional disulfide linkages between opposing F(ab') arms through a structure-guided disulfide engineering strategy. This resulted in rigid and compact molecules with significantly enhanced agonistic activity, compared to the parental antibody. Our study generalizes the concept of disulfide engineering and supports rational design to further augment the biological activity of therapeutic antibodies targeted against this receptor class.

## Results

### Modification of hinge disulfides similarly affects agonistic activity and flexibility in anti-hCD40 and anti-h4-1BB hIgG2 mAb

Since previous work showed that stronger agonism of the anti-hCD40 hIgG2 mAb ChiLob7/4 was associated with a cross-over of disulfides in the hinge region that restricted antibody conformation[20], we investigated here whether this effect was generalizable to other immunostimulatory antibodies. We constructed equivalent C-S exchange variants in the anti-h4-1BB mAb SAP1.3, as previously described for the anti-hCD40 mAb ChiLob7/4[20]. We introduced the most flexible (C232S + C233S), the most rigid (C232S κC214S), and an intermediate (C233S κC214S) variant into the hIgG2 framework, alongside hIgG1 (−) and hIgG2 (+) controls (Fig. 1a). For comparison we also produced new preparations of the same ChiLob7/4 variants. The antibodies produced well, with typical IgG properties and low aggregation, as determined by HPLC (Supplementary Fig. 1 and Supplementary Table 1). All mAb were characterized using reducing and non-reducing SDS-PAGE and capillary electrophoresis with sodium dodecyl sulfate (CE-SDS) analysis to confirm correct formation of the molecules (Supplementary Fig. 2). Native wild-type (WT) hIgG1 and hIgG2, as well as the hIgG2 C232S + C233S variant were observed as a single species migrating at ~150 kDa under non-reducing conditions. However, the C233S κC214S and C232S κC214S hIgG2 variants exhibited two species at ~23 and 104 kDa (Supplementary Fig. 2a) due to dissociation of the light chains from the heavy chain dimeric complex. This dissociation is suggestive of the formation of the C127-C233 or C127-C232 disulfide cross-overs, leading to a lack of a stabilizing disulfide between the heavy and light chains (C127-κC214) (Fig. 1a). The mAb displayed thermal stability equivalent to the parental hIgG2 wild-type (Supplementary Fig. 3 and Supplementary Table 2), indicating that structural integrity and stability were not compromised.

The agonistic activity of the mAb variants was then assessed using an NF-κB/Jurkat/GFP reporter cell line transfected with either hCD40 or h4-1BB, where GFP is produced in response to NF-κB activation following effective receptor clustering[23]. The same trends in activity were seen for the hIgG2 C-S variants in both anti-hCD40 ChiLob7/4 and anti-h4-1BB SAP1.3, with C232S κC214S the most active variant and C232S + C233S the least active, displaying similar levels of activity to the WT hIgG1 (Fig. 1b).

To assess if the C-S exchanges altered receptor binding and affinity, surface plasmon resonance (SPR) and binding assays with h4-1BB or hCD40-expressing Jurkat cells were performed. The SPR data showed that the C-S variants retained high affinity binding to their respective receptors ($K_D$ range between 4.9 and 13.8 nM for SAP1.3 hIgG2 variants and between 0.6 and 6.7 nM for ChiLob7/4 hIgG2 variants) (Supplementary Table 3 and Supplementary Fig. 4). Similar binding to the relevant h4-1BB or hCD40-expressing cells was observed (Supplementary Fig. 5), with differences consistent with receptor distribution changes induced by different variants at the cell surface as described previously[23]. These observations are in line with our previous studies showing that high-affinity binding is retained for C-S exchange variants[20], owing to all mAb having identical variable regions.

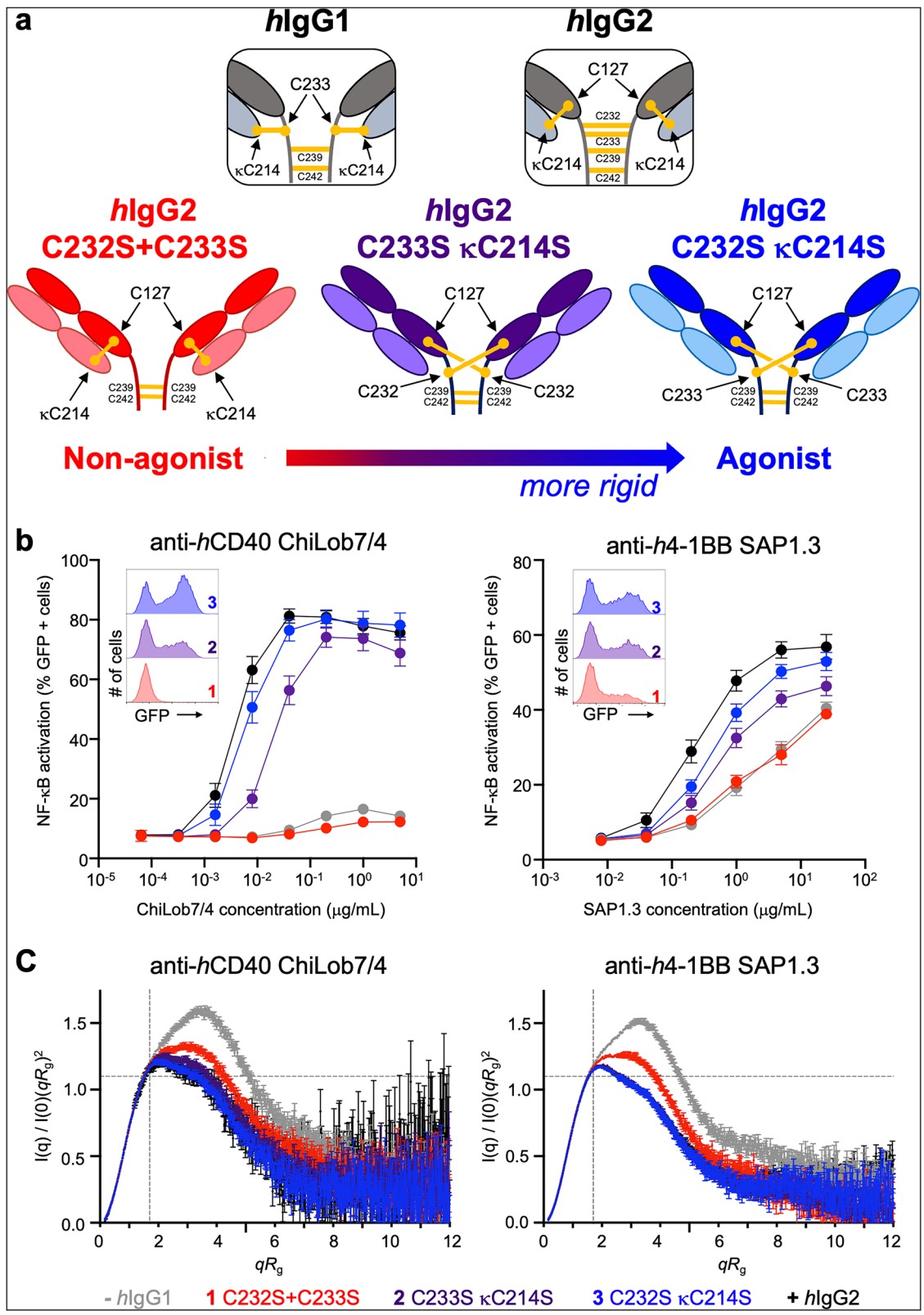

To understand the effects of the C-S mutations on conformation, we used SAXS to evaluate conformational states in solution. As $h$IgG2 activity is independent of the Fc[18,19,22], F(ab')$_2$ fragments were produced by pepsin digestion for SAXS to remove the confounding complexity of the flexible Fc region. Primary analysis of the SAXS data for both the anti-$h$CD40 and anti-$h$4-1BB F(ab')$_2$ fragments showed the same trends in the radius of gyration (R$_g$) and maximum intramolecular distance

(D$_{max}$), which corresponded with activity (Supplementary Figs. 6, 7 and Supplementary Table 4, 5). The distance distribution function (P(r)) analysis (Supplementary Fig. 8) showed that the most agonistic variant, $h$IgG2 C232S κC214S, displayed the smallest R$_g$ and D$_{max}$ values for both ChiLob7/4 (R$_g$ = 40.00 ± 0.12 Å, D$_{max}$ = 139.0 Å) and SAP1.3 (R$_g$ = 39.12 ± 0.06 Å, D$_{max}$ = 129 Å), whilst the least agonistic variant, $h$IgG2 C232S + C233S, displayed the largest R$_g$ and D$_{max}$ values for both

**Fig. 1 | C-S disulfide variants exhibit the same trend in agonistic activity and conformation in anti-$h$CD40 and anti-$h$4-1BB mAb. a** Model of disulfide arrangements in $h$IgG2 mAb affecting agonistic activity and flexibility, derived from studies of the $h$IgG2 anti-$h$CD40 ChiLob7/4[20]. Disulfides shown in yellow, and cysteines involved in disulfide bonds labeled (with the Kabat numbering system). $h$IgG1 and $h$IgG2 wildtypes are shown inset above. **b** NF-κB/Jurkat/GFP reporter cells expressing $h$CD40 or $h$4-1BB were stimulated with serially diluted ChiLob7/4 or SAP1.3 mAb, respectively. NF-κB activation triggers GFP expression and was quantified after 24 h by flow cytometry (inset histograms representative from $n = 3$ at 0.04 μg/ml for ChiLob7/4 and 5 μg/mL for SAP1.3). Graphs show dose-response

curves of the percentage of GFP-positive cells. $n = 3$–7 independent biological experiments; mean ± SEM, taken from technical triplicates for each independent experiment. $h$IgG1 (gray) and $h$IgG2 (black) shown as controls. **c** F(ab')$_2$ of anti-$h$CD40 and anti-$h$4-1BB C-S disulfide variants were analyzed by SEC-SAXS. Graphs show dimensionless Kratky plots derived from SEC-SAXS data, with the Guinier-Kratky point ($\sqrt{3}$, 1.103) indicated by the pale gray crosshairs. $h$IgG1 (gray) and $h$IgG2 (black) shown as controls. Errors calculated following standard BioXTAS RAW software procedures[63]. Disulfide C-S antibody variants labeled by color: red C232S + C233S, purple C233S κC214S, blue C232S κC214S. Source data are provided as a Source Data file.

ChiLob7/4 ($R_g = 43.56 \pm 0.10$ Å, $D_{max} = 148$ Å) and SAP1.3 ($R_g = 43.36 \pm 0.09$ Å, $D_{max} = 151$ Å). These data are consistent with a more compact structure for the agonistic variants.

Conformation was further evaluated using dimensionless Kratky plots. A perfectly globular protein would give a Gaussian-like curve peaking at the Guinier-Kratky point, shown as the gray crosshair (Fig. 1c), and deviation from this distribution is indicative of either elongation of the particle or increased flexibility. For both ChiLob7/4 and SAP1.3, all variants deviated from a Gaussian distribution, typical for an F(ab')$_2$ fragment due to its non-globular shape. Of the C-S exchange variants, C232S κC214S displayed the least deviation from a Gaussian distribution, whilst C232S + C233S showed the greatest deviation in both ChiLob7/4 and SAP1.3 formats. The WT $h$IgG1, which is least agonistic, showed the greatest deviation, as has been seen previously[29–31]. Together these data show that the more agonistic C-S variants are associated with a more compact conformation compared to the less agonistic C-S variants.

SEC-SAXS data for the ChiLob7/4 and SAP1.3 variants as IgG displayed the same trends in $R_g$, $D_{max}$, and Kratky plots as the F(ab')$_2$ fragments (i.e., C232S κC214S was the least deviated from a Gaussian distribution and C232S + C233S the most), indicating that the conformational differences are retained in whole IgG, with the observed effects independent of the Fc (Supplementary Figs. 9, 10, Supplementary Table 6, 7).

## Structure-guided disulfide design in $h$IgG2 mAb

Next, we sought to investigate whether further restriction in conformation, through the introduction of additional disulfides between opposing F(ab') arms, using rational engineering approaches, would result in yet further increased agonistic activity. Here, we focussed our efforts on design using structural insights from the anti-$h$CD40 mAb ChiLob7/4, exploiting the previously determined crystal structure of the most agonistic disulfide cross-over variant $h$IgG2 C232S κC214S (PDB: 6TKE) as a framework on which to introduce further mutations (Fig. 2a and Supplementary Fig. 11). The C232S κC214S variant is henceforth referred to as cross-over. Two different structure-guided approaches were used to engineer additional disulfide bonds inside or outside of the $h$IgG2 hinge.

Visual inspection of the crystal structure of the upper hinge aimed to identify areas where opposing heavy chains were close enough to introduce a pair of cysteines that would lead to the formation of a disulfide bond to link F(ab') arms (Supplementary Fig. 11a). The amino acid K228 was deemed to be in a suitable position to target for mutation to cysteine, based on distance measurements. MODELER[32] was used to generate a predicted structure of ChiLob7/4 $h$IgG2 C232S κC214S (cross-over), with the additional K228C mutation, to determine whether a disulfide bond between the amino acid 228 on opposing heavy chains would be likely to form (Fig. 2b, left panel, Supplementary Fig. 11b). With the assumption that the original hinge disulfide cross-over is maintained, the C232S + K228C κC214S variant was named cross-over + K228C.

In an alternative approach, we investigated the introduction of mutations outside the hinge using the disulfide engineering software

*Disulfide by Design 2.0*[33]. The software predicts pairs of amino acids, based on a given structure, that will likely form disulfides if mutated to cysteine, and identified the amino acids T222 and κE123 on opposing heavy and light chains outside the hinge (Fig. 2b, right panel, Supplementary Fig. 11c). Again, assuming the hinge disulfide cross-over is maintained, the C232S + T222C κE123C + κC214S variant was named cross-over + T222C κE123C.

The new mAb variants were produced and characterized by SDS-PAGE and CE-SDS (Supplementary Fig. 12) and retained an equivalent thermal stability profile to the parental cross-over variant (Supplementary Fig. 3, Supplementary Table 2). As seen with the parental cross-over variant (Supplementary Fig. 2), the cross-over + K228C variant showed two species migrating at ~23 kDa and ~104 kDa, indicating dissociation of the light chains from the heavy chain dimer (Supplementary Fig. 12). This would occur if, as predicted, these variants lack the disulfide that links the light and heavy chains, between C127-κC214, due to the disulfide cross-over between C127-C233. The cross-over + T222C κE123C variant migrated as a single species at ~150 kDa, suggesting the presence of additional disulfides linking heavy and light chains together.

We, therefore, continued with crystallographic structure determination to ascertain the disulfide topology (Supplementary Table 8). X-ray crystal structures were determined (Fig. 2c) from pepsin-digested F(ab')$_2$ fragments (Supplementary Fig. 13). Sulfur positions were determined using an anomalous scattering approach, sulfur single-wavelength anomalous dispersion (Sulfur SAD) (Fig. 2d). Anomalous electron density confirmed that the structure-guided design process was successful, with disulfides forming as intended. In the cross-over + K228C variant, the disulfide cross-over between C127 and C233 formed in addition to a C228-C228 disulfide on opposing heavy chains. In the cross-over + T222C κE123C variant, the disulfide cross-over formed (C127-C233) in addition to two disulfides (C222-κC123) linking opposing heavy and light chains on opposite F(ab') arms. This observation readily explains why the light and heavy chains did not dissociate on SDS-PAGE or CE-SDS (Supplementary Figs. 12, 13) despite the fact that the C127-κC214 disulfide, which would normally link heavy and light chains, cannot form.

## Disulfide engineering results in rigid and compact antibody molecules

We went on to characterize conformational restriction using SAXS combined with MD simulations for the F(ab')$_2$ fragments, as before[20]. Primary analysis of the SAXS data indicated that both designed variants were slightly more compact than the parental cross-over in solution, with decreased $R_g$ values in the distance distribution function P(r) (cross-over + K228C variant, $R_g = 39.45 \pm 0.11$ Å and cross-over + T222C κE123C variant, $R_g = 39.22 \pm 0.10$ Å, compared to the parental cross-over, $R_g = 40.00 \pm 0.12$ Å) (Table 1, Supplementary Table 4 and Supplementary Fig. 6). The same trends in $R_g$ and $D_{max}$ are seen for the variants when characterized as IgG rather than F(ab')$_2$ (Supplementary Table 6 and Supplementary Fig. 9), demonstrating that the conformational differences are retained in whole IgG and are independent of the Fc region.

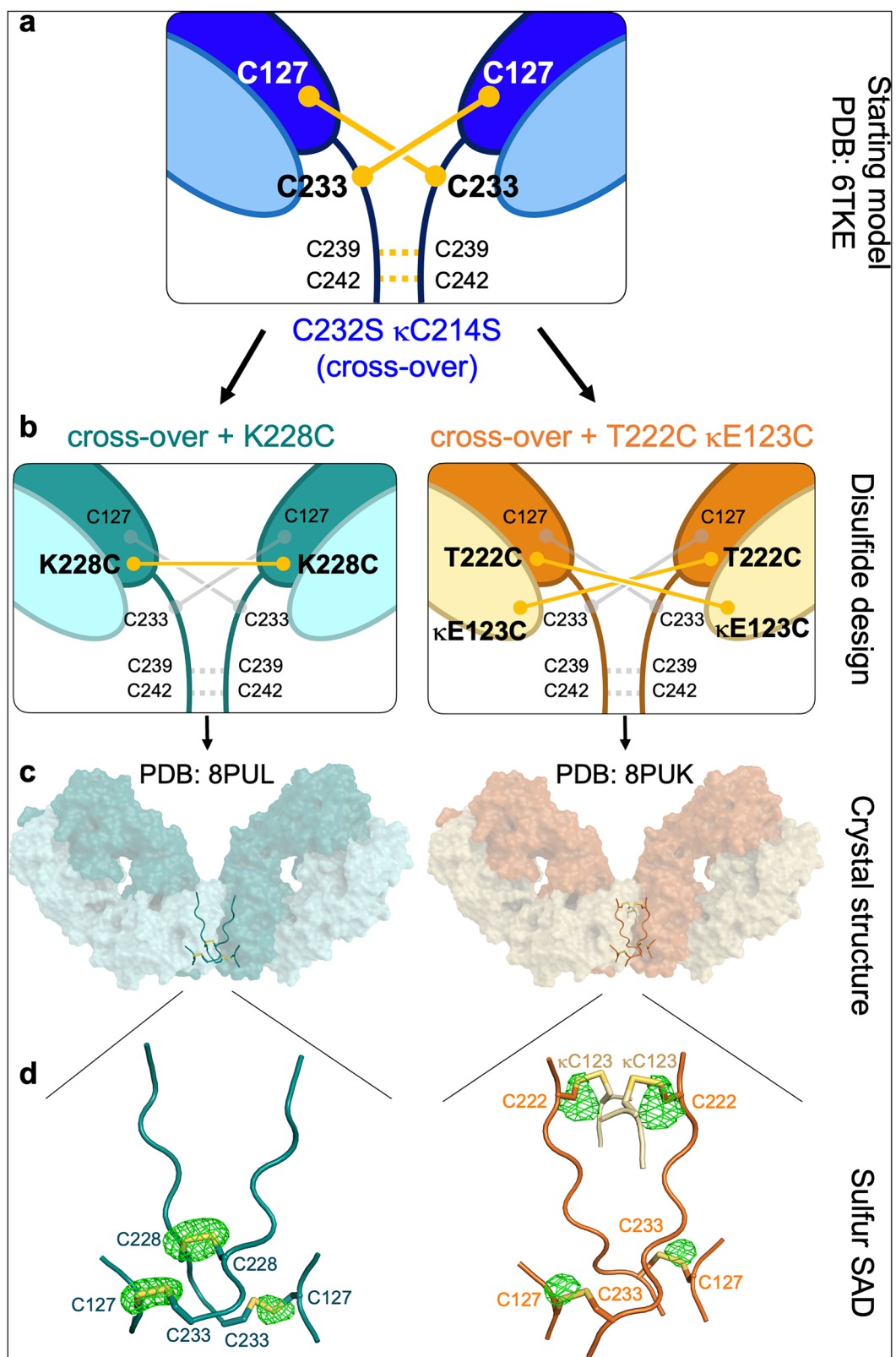

The dimensionless Kratky plots suggest subtle differences between the engineered variants and the parental cross-over (Fig. 3a, Supplementary Fig. 14a). Therefore, to explore whether the variation in disulfide pattern, as seen in the crystal structures (Fig. 2c, d), would lead to discernible conformational differences, we used atomistic MD simulations to generate structural states for each variant from three replicate trajectories of 2 µs. The theoretical scattering curves were calculated for each of the atomistic models within the MD-generated conformational pools, and these curves were compared to the corresponding experimental scattering data to find the best-fitting single models. For the parental cross-over variant and the two engineered variants, the best-fitting single models showed good agreement with the SAXS data (Fig. 3b and Supplementary Table 4), with $\chi^2$ values of 1.081 for the parent, 1.311 for the cross-over + K228C variant and 1.009

**Fig. 2 | Structure-guided design of anti-*h*CD40 *h*IgG2 mAb with additional engineered disulfides. a** Schematic of the parental agonistic *h*IgG2 C232S κC214S variant (cross-over) showing experimentally determined disulfides as solid yellow lines. Disulfides not resolved in the structure (PDB: 6TKE) are shown as dashed yellow lines. Cysteine amino acid residues are labeled. **b** Structure-based design and predicted disulfides for C232S + K228C κC214S (cross-over + K228C) and C232S + T222C κE123C + κC214S (cross-over + T222C κE123C) shown in yellow, with disulfides from the parent cross-over variant in gray. **c** Experimentally determined crystal structures of the new F(ab')$_2$ variants shown as surface representation, with disulfides as sticks. **d** Sulfur single-wavelength anomalous dispersion (Sulfur SAD) crystallography reveals the position of sulfur atoms and confirms disulfides between neighboring chains. The anomalous electron density is shown as green mesh (anomalous difference Fourier map, contoured at 5 σ). Disulfides shown in yellow as sticks. Engineered antibody variants labeled by color: blue C232S κC214S (cross-over), teal cross-over + K228C, orange cross-over + T222C κE123C.

**Table 1 | Restriction in conformation of new anti-*h*CD40 variants revealed using SAXS analysis**

|  | C232S + C233S | C232S κC214S (cross-over) | cross-over + K228C | cross-over + T222C κE123C |
|---|---|---|---|---|
| $R_g$ (Å) | 43.56 ± 0.10 | 40.00 ± 0.12 | 39.45 ± 0.11 | 39.22 ± 0.10 |
| $D_{max}$ (Å) | 148 | 139 | 132 | 132 |

F(ab')$_2$ were analyzed by SEC-SAXS. Shown are the experimentally determined $R_g$ and $D_{max}$ values derived from P(r) analysis.

for the cross-over + T222C κE123C. These data suggested that our disulfide designs yielded antibody molecules which were conformationally consistent with the parent cross-over molecule, as can be seen from the structures of the best-fitting models (Fig. 3c). The good agreement of individual models extracted from the MD-generated conformational pools indicates that the engineered variants, as well as the parent, predominantly adopt a single conformation or a highly restricted conformational ensemble under the conditions tested, allowing the SAXS data to be described well by a single representative structure. In contrast, the best-fitting single models for C232S + C233S and C233S κC214S showed poorer agreement with the SAXS data, with substantial structure seen in the error-weighted residuals and thus required ensemble fitting (Supplementary Methods, Supplementary Fig. 15).

Whilst the two engineered variants appear structurally similar to the parent cross-over, there are subtle differences in conformation observed in the hinge angles (calculated according to Supplementary Fig. 16), $R_g$ and $D_{max}$ of the best-fitting single models (Fig. 3d–f and Supplementary Table 4). The designed variants are slightly more compact with hinge angles of 125.3° for the cross-over + K228C and 124.9° for the cross-over + T222C κE123C, compared to a hinge angle of 126.6° for the parent. Similarly, the $R_g$ for the best fitting model for each of the designed variants is smaller (38.93 Å for the cross-over + K228C and 39.07 Å for the cross-over + T222C κE123C, compared to 39.57 Å for the parent). Together, our observations support that our engineered disulfides result in rigid and compact antibody molecules, similar to the parental cross-over variant.

### Disulfide-engineered mAb variants have greater agonistic activity

We hypothesized that the disulfide-engineered mAb variants would provide greater agonistic activity than the more flexible variants. The engineered variants were thus assessed for receptor binding and biological activity. We first used SPR and cell binding assays as before[20] to confirm that the designed variants still bound *h*CD40 with high affinity (Fig. 4a, Supplementary Table 9 and Supplementary Figs. 17, 18).

To investigate the agonistic activity, the *h*CD40-expressing NF-κB/Jurkat/GFP reporter cell line was used, as described previously (Fig. 1b). At the highest concentrations, the designed variants had similar levels of activity to the parental cross-over (Fig. 4b). However, at lower concentrations, the designed variants had significantly higher levels of NF-κB activation compared to the parent (*p* < 0.01 at 0. 008 μg/mL, see Supplementary Table 10). Both variants exhibited ~3 fold lower EC$_{50}$ values ($1.55 \times 10^{-3}$ μg/mL for both the cross-over + T222C κE123C and the cross-over + K228C) for activity compared to the parental cross-over ($4.59 \times 10^{-3}$ μg/mL) (Table 2).

We then used primary B cells purified from human peripheral blood mononuclear cells (PBMCs) to provide a more physiologically relevant readout to evaluate agonistic activity. We assessed CD40 agonism by measuring homotypic adhesion (cell:cell clustering), upregulation of CD23, CD86, and HLA-DR and B cell proliferation, using assays which we have previously shown to correlate well with therapeutic impact[18,19,22]. Homotypic adhesion (indicated by darker regions in Fig. 4c) was greatest for the two new engineered variants. Flow cytometric analysis showed that HLA-DR, CD23, and CD86 upregulation were significantly greater for the cross-over + K228C and the cross-over + T222C κE123C compared to the parent cross-over (Fig. 4d–f and Supplementary Table 11), as was B cell proliferation, measured as an increase in ³H-thymidine incorporation (Fig. 4g and Supplementary Table 11). As seen previously[20], the *h*IgG1 wildtype was agonistically inert, comparable to the isotype controls. The same trends in agonistic activity were seen using F(ab')$_2$ fragments rather than full IgG molecules, indicating that the agonistic activity seen is Fc-independent (Supplementary Fig. 19 and Supplementary Table 12). Given this consistency across four different cellular read-outs, we conclude that engineering additional disulfides to link opposing antibody F(ab') arms results in a compact and rigid antibody molecule with augmented biological activity.

## Discussion

In our study, we investigated whether the association between agonism and conformation observed previously in the anti-*h*CD40 *h*IgG2 mAb ChiLob7/4 was generalizable to another antibody targeting a different co-stimulatory receptor. We then explored whether designing additional linkages between opposing F(ab') arms in the rigid *h*IgG2 antibody variant C232S κC214S (termed cross-over) provided stronger agonism.

Naturally occurring *h*IgG isotypes have a characteristic hinge length and disulfide pattern, with an established hierarchy of flexibility from most to least flexible of *h*IgG3 > *h*IgG1 > *h*IgG4 > *h*IgG2[34]. We and others have shown for several anti-*h*CD40 mAb that conformationally diverse antibody species are associated with decreased agonism, with *h*IgG1 conformationally diverse and least active and *h*IgG2 the most rigid and most active[19,35].

We demonstrate that introducing C-S exchange mutations into the hinge of *h*IgG2 mAb targeting the co-stimulatory receptors *h*CD40 and *h*4-1BB modifies conformational freedom, with consequent effects on agonistic activity. We observe a consistent trend in activity for anti-*h*CD40 ChiLob7/4 and anti-*h*4-1BB SAP1.3, with the *h*IgG2 C232S + C233S variant showing the lowest levels of agonism, and the *h*IgG2 C232S κC214S variant exhibiting the strongest agonism. The most

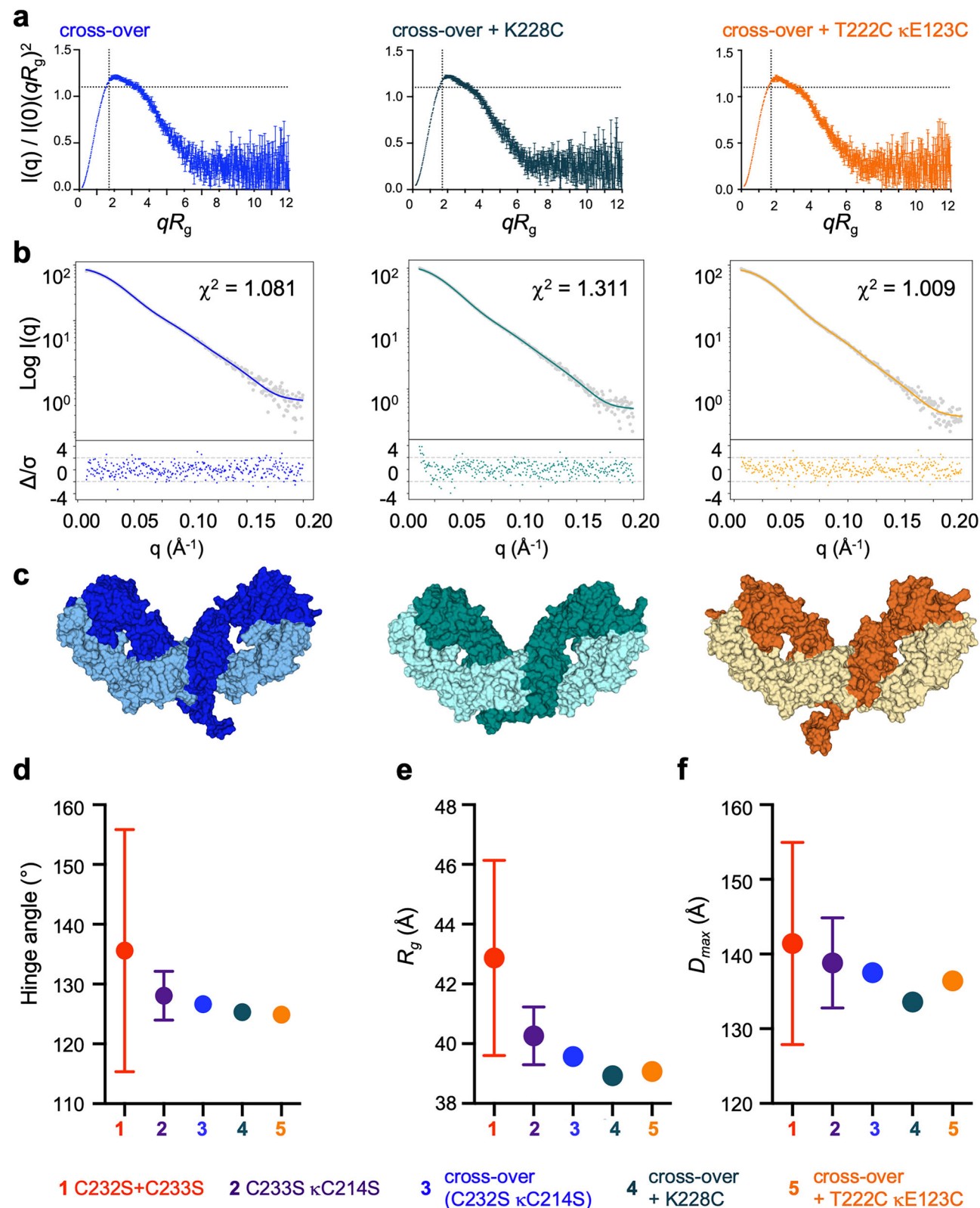

**d 1** C232S+C233S    **2** C233S κC214S    **3** cross-over (C232S κC214S)    **4** cross-over + K228C    **5** cross-over + T222C κE123C

agonistic variant, C232S κC214S was previously characterized in Chi-Lob7/4 to contain a disulfide cross-over, stabilizing the hinge region[20]. Similar solution-phase properties were seen with anti-*h*CD40 ChiLob7/4 and anti-*h*4-1BB SAP1.3 F(ab')₂ fragments for each of the variants, with *h*IgG2 C232S + C233S displaying more diverse conformations, and *h*IgG2 C232S κC214S adopting a more compact molecular arrangement.

Whilst all the *h*IgG2 disulfide variants retained high affinity binding, differences were seen in the actual values. The differences in affinity were largely due to variations in the off-rate and were more significant for ChiLob7/4 than for SAP1.3. Moreover, differences were also observed when comparing native *h*IgG1 versus *h*IgG2. Together, these data suggest contributions from the F(ab') and hinge regions in this effect.

**Fig. 3 | Conformationally rigid and compact nature of engineered anti-*h*CD40 variants revealed through SAXS fits to models extracted from MD simulations.** F(ab')$_2$ were analyzed by SEC-SAXS. **a** Graphs show dimensionless Kratky plots derived from the SEC-SAXS, with the Guinier-Kratky point (√3, 1.103) indicated by the pale gray crosshairs. Errors calculated following standard BioXTAS RAW software procedures[63]. **b** Models for each F(ab')$_2$ were extracted from 6 μs of MD simulation, every 1 ns. Agreement of the calculated scattering curves for the extracted models to the experimental scattering data were calculated using CRYSOL, with the best fitting single model shown (calculated scattering shown in color, experimental scattering shown as gray dots). $x^2$ fit with error-weighted residuals plot also shown. **c** Conformation of the best fitting single model shown, with heavy chain in a darker shade, light chain in a lighter shade. **d** For variants 3–5, the hinge angle for best fitting single model shown. For variants 1-2, the mean hinge angle (± SD) for the best-fitting GAJOE-selected ensemble shown. **e** For variants 3–5, $R_g$ for best fitting single model shown. For variants 1-2, mean $R_g$ (±SD) for the best fitting GAJOE-selected ensemble shown. See Supplementary Fig. 15c. **f** For variants 3–5, $D_{max}$ for best fitting single model shown. For variants 1-2, the mean $D_{max}$ (± SD) for the best-fitting GAJOE-selected ensemble shown. See Supplementary Fig. 15d. Engineered antibody variants labeled by color: red C232S + C233S, purple C233S κC214S, blue C232S κC214S (cross-over), teal cross-over + K228C, orange cross-over + T222C κE123C. Source data are provided as a Source Data file.

We further showed that it is possible to engineer disulfides into the *h*IgG2 framework using structure-guided approaches, resulting in two new *h*IgG2 antibody molecules, each with different additional linkages between opposing F(ab') arms. We used a combination of design informed by structure inspection, as well as a software-led approach, to identify amino acids to change to cysteine both within and outside of the hinge. We demonstrate that the designed disulfides result in a rigid conformation similar to the parent molecule C232S κC214S (cross-over). While the new variants are slightly more compact than the parent, these differences were subtle and difficult to evaluate with the current methodology. Despite this, we show that even small structural and conformational differences in these molecules can elicit substantial differences in agonistic activity, with the two engineered variants exhibiting significantly greater biological activity than the parent. This increased agonism was demonstrated through NF-κB reporter assays, homotypic adhesion, elevated B cell activation, and increased B cell proliferation, providing a consistent elevation in activity.

Future design of antibodies must consider the mechanism by which agonism is achieved. Conformational restriction likely impacts on biological activity by modulating receptor clustering. Agonistic activity is directly associated with receptor clustering for both anti-*h*CD40 and anti-*h*4-1BB mAb[23]. The restricted conformational diversity of disulfide-engineered variants may retain receptors in closer proximity, promoting efficient clustering of receptors in the cell membrane trigger activation of downstream intracellular signaling pathways, thus leading to cellular activation. In contrast, mAb variants with greater conformational freedom would be less likely to stabilize receptor clusters and thus be incapable of surpassing the receptor signaling threshold to promote strong agonism.

Regulation of agonism by *h*IgG antibodies and the consequent differences in conformational rigidity is not limited to the anti-*h*CD40 mAb ChiLob7/4 and the anti-*h*4-1BB mAb SAP1.3. The impact of *h*IgG2 on promoting receptor clustering to provide strong agonism in comparison to *h*IgG1 has been shown for mAb targeting different epitopes of multiple different TNFRs, including the clinically relevant anti-*h*CD40 mAb 341G2 and SGN40[19,22], the clinically-relevant anti-*h*4-1BB mAb Urelumab, the anti-*h*OX40 mAb SAP9[23], the anti-*h*CD27 mAb varlilumab, hCD27.15, AT133-2 and AT133-11[12] and several anti-*h*DR5 mAb[35]. This principle also extends to mAb targeting receptors from different receptor families such as immunoglobulin superfamily member CD28[19] and also the CD200 receptor[36].

Alternative methods for engineering conformational restriction of mAb include the recently reported i-shaped antibody approach[37]. By utilizing intramolecular Fab-Fab homotypic interfaces, the typical Y-shaped antibody structure was transformed into a more compact and constrained i-shape. This concept was shown to increase agonistic activity for a number of mAb targeting TNFRSF members, including CD40, 4-1BB, DR4, and DR5, highlighting the potential translatability of an engineering approach which aims to restrict conformation to induce greater agonism. Similarly, the Contorsbody format, which converts a typical IgG into a geometrically altered, sterically constrained format, has greater rigidity and agonistic potential compared to canonical antibodies[38].

The concept of conformational restriction could also be applied to multi-specific mAb such as those targeting co-stimulatory receptors with one or more F(ab') arm and a tumor-associated antigen with the other F(ab') arm. By utilizing our disulfide engineering approaches to reduce the flexibility of multi-specific mAb, it may be possible to bring the tumor into considerably closer proximity to the costimulatory receptor-expressing T cells than previously seen, thus providing more directed immune activation.

This study uses disulfide engineering approaches to illustrate the importance of structure and conformation in the development of therapeutic agonistic mAb. We show that modifying the conformation and rigidity of mAb through the manipulation and introduction of disulfides both within and outside the hinge region has significant effects on biological activity and function. These methods represent powerful tools to tune the activity of agonist antibodies, which could be broadly applicable in the design and optimization of biotherapeutics targeting a range of receptor classes. This should enable the development of more effective mAb for clinical use.

## Methods

Research performed in this study complies with all relevant ethical regulations of the University of Southampton; with human samples assessed via the Faculty of Medicine Research Ethics Committee under submission 19660. Blood cones were obtained from healthy adult donors through Southampton National Blood Services with prior informed consent. The use of human blood for these assays was approved by the East of Scotland Research Ethics Service, Tayside, UK.

### Antibody production and purification

The *h*IgG2 disulfide variants (C232S + C233S, C233S κC214S, C232S κC214S (cross-over), C232S + K228C κC214S (cross-over + K228C) and C232S + T222C κE123C + κC214S (cross-over + T222C κE123C) were generated in ChiLob7/4 or SAP1.3 using site-directed mutagenesis. The Kabat numbering scheme was used throughout but sequential numbering was used in the deposition of the structures in the PDB. Accordingly, the Kabat amino acids C127, T222, K228, C232, C233, C239 and C242 correspond respectively to C136, T219, K223, C224, C225, C228 and C231 in the deposited models of anti-*h*CD40 ChiLob7/4. For anti-*h*4-1BB SAP1.3, the Kabat amino acids C232 and C233 correspond to C226 and C227, respectively. The κ light chain amino acids, κE123 and κC214, are identical in both nomenclatures.

Antibodies were produced in ExpiCHO-S cells using the Gibco ExpiCHO transient expression system (ThermoFisher UK) and purified by protein A affinity chromatography using a HiTrap MAbSelect SuRe protein A column (Cytiva), followed by size exclusion chromatography (SEC) using a HiLoad Superdex 200 pg 16/600 SEC column (Cytiva), if required. All antibodies were determined to be endotoxin low (< 1 EU/mg) using an Endosafe Portable Test System device (Charles River Laboratories) and aggregate-free (< 1%) using high-performance liquid chromatography (HPLC). For samples > 1% aggregation, caution was taken with interpretation. Antibody titer and final IgG purity are

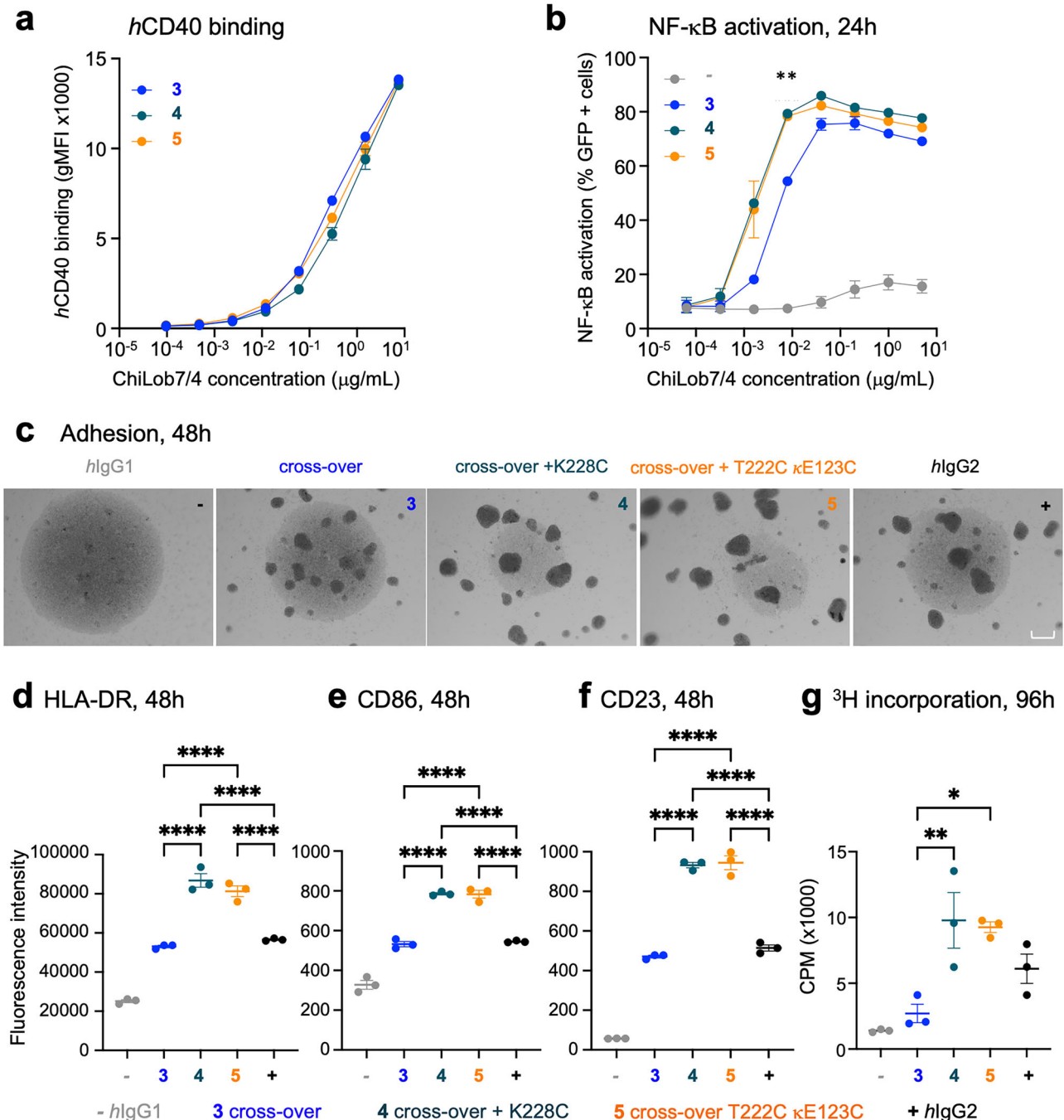

**Fig. 4 | Receptor binding and agonistic activity of engineered anti-$h$CD40 mAb. a** Serially diluted ChiLob7/4 variants were incubated with Jurkat cells expressing $h$CD40 with binding detected with a secondary PE-conjugated antibody by flow cytometry. Graphs show binding dose-response curves of geometric mean fluorescence intensity (gMFI). Mean ± SEM, $n = 2$ independent biological experiments, mean taken from technical triplicate for each independent experiment. **b** NF-κB/Jurkat/GFP reporter cells expressing $h$CD40 were stimulated with serially diluted ChiLob7/4 variants, and activation quantified after 24 h by determination of GFP expression levels, assessed by flow cytometry. The graph shows dose-response curves of percentage GFP + cells for the variants. Mean ± SEM, $n = 2$ independent biological experiments, mean taken from technical triplicate for each independent experiment. ** $p < 0.01$ at 0.008 μg/mL, one-way ANOVA with Tukey's multiple comparisons tests (for $p$-values and significance levels for other concentrations, see Supplementary Table 10). **c–g** ChiLob7/4 variants were incubated with human B cells, and various activation assays performed. **c** Homotypic adhesion of human B cells measured 48 h after addition of 0.008 μg/mL ChiLob7/4 mAb. Images are representative of technical triplicates from 1 of 3 independent experiments. Scale bar 200 μm. **d–f** Activation of primary human B cells determined by upregulation of (**d**) HLA-DR, (**e**) CD86, (**f**) CD23, using flow cytometry; measurements taken 48 h after addition of 0.008 μg/mL ChiLob7/4 mAb. **g** Proliferation of human B cells assessed by $^{3}$H-thymidine incorporation 96 h after addition of 0.008 μg/mL ChiLob7/4 mAb. CPM = counts per minute. *$p < 0.05$, **$p < 0.01$, ***$p < 0.001$, ****$p < 0.0001$ one-way ANOVA with Tukey's multiple comparisons tests (for exact p-values, see Supplementary Table 11). For c, data show representative images from technical triplicate from 1 of 3 independent experiments with independent donors. For d-g, data show technical triplicates (mean ± SEM) from 1 of 3 independent experiments with independent donors. Engineered antibody variants labeled by color: blue C232S κC214 (cross-over), teal cross-over + K228C, orange cross-over + T222C κE123C. $h$IgG1 (gray) and $h$IgG2 (black) shown as controls. Source data are provided as a Source Data file.

**Table 2 | EC$_{50}$ of NF-κB activation**

| Antibody | EC$_{50}$ (x10$^{-3}$ µg/mL) |
|---|---|
| C232S κC214S (cross-over) | 4.59 ± 0.142 |
| cross-over + K228C | 1.55 ± 0.0311 |
| cross-over + T222C κE123C | 1.55 ± 0.566 |

NF-κB/Jurkat/GFP reporter cells expressing $h$CD40 were stimulated with serially diluted Chi-Lob7/4 variants, and activation quantified after 24 h by determination of GFP expression levels, assessed by flow cytometry. Dose-response curves (see Fig. 4b) were analyzed to determine the half-maximal effective concentration to induce NF-κB activation, $n$ = 2 independent experiments.

shown in Supplementary Table 1, and HPLC traces are shown in Supplementary Fig. 1.

F(ab')$_2$ fragments were generated by digesting IgG with pepsin (Sigma) for 1-2 h at 37 °C to remove the Fc domain, with the regular observation of digestion progress by HPLC. F(ab')$_2$ fragments were purified by gel filtration using a HiLoad Superdex 200 pg 16/600 SEC column (Cytiva) to remove undigested IgG, followed by further purification of the pooled fractions using a HiTrap MabSelect SuRe protein A column (Cytiva) to remove any residual IgG and Fc. Purified F(ab')$_2$ fragments were checked by HPLC to confirm purity and then concentrated using Amicon Ultra Centrifugal filters (Millipore, Sigma Aldrich) with a 10,000 kDa cutoff.

Antibodies used in this study were produced in mammalian CHO cells, resulting in Fc glycosylation. This occurs in all antibodies (including natural antibodies produced in vivo) at the N297 residue in the Fc domain. Glycosylation was not removed prior to our biological assays or IgG SAXS data collection. Glycosylation can also occur in the F(ab') regions. ChiLob7/4 has no glycosylation sites in its F(ab') regions, so the ChiLob7/4 F(ab')$_2$ fragments do not include glycosylation, whereas SAP1.3 does have a glycosylation site in its F(ab') region so the SAP1.3 F(ab')$_2$ fragments may include glycosylation.

## Assessment of immunostimulatory activity

The immunostimulatory activity was evaluated using the NF-κB/Jurkat/GFP Transcriptional Reporter cell line (System Biosciences, USA), expressing either full-length $h$CD40 or $h$4-1BB extracellular domain with the $h$CD40 intracellular cytoplasmic tail, as described previously[23]. Briefly, Jurkat cells were incubated with serially diluted mAb for 24 h at 37 °C. The degree of NF-κB activation was quantified by measuring GFP fluorescence using flow cytometry.

The immunostimulatory activity of anti-$h$CD40 mAb was also assessed using primary human B cells purified from human peripheral blood mononuclear cells (PBMCs). PBMCs were isolated from fresh leukocyte cones by density gradient centrifugation. Human B cells were purified from PBMCs by negative selection using a MojoSort Human B cell Isolation kit (BioLegend). B cells were incubated in vitro with anti-$h$CD40 mAb or F(ab')$_2$ in 96 well round-bottom plates.

To measure homotypic adhesion, B cells were imaged 48 h after the addition of mAb with a conventional light microscope (Olympus CKX41, running Olympus CellSens Standard software). Adhesion was observed as large macroscopic cell groupings. Upregulation of B cell activation marker expression was assessed by flow cytometry after 48 h, using APC-labeled anti-CD23 mAb (1/100, clone EBVCS-5, BioLegend), PerCP-Cy5.5-labeled anti-CD86 mAb (1/200, clone BU63, BioLegend) and Brilliant Violet-labeled anti-MHCII mAb (1/100 clone L243, BioLegend). To assess B cell proliferation, B cells were stimulated with mAb or F(ab')$_2$ as above for 4 days, with 1 µCi of $^3$H−thymidine (PerkinElmer) added to each well for the last 18 h of incubation. Cells were harvested and analyzed by scintillation counting (TopCount) to measure $^3$H-thymidine incorporation.

## Assessment of antibody cell surface receptor binding

The binding of mAb to cell surface receptors was assessed using Jurkat cells expressing the relevant receptor. Jurkat cells were incubated with serially diluted mAb for 30 mins at 4 °C. Cells were then washed twice in buffer containing phosphate-buffered saline, 1 % bovine serum albumin, and 0.01% sodium azide by centrifugation. The remaining $h$IgG was detected by incubation with a secondary PE-conjugated polyclonal goat F(ab')$_2$ anti-$h$IgG Fc-specific antibody (Jackson ImmunoResearch Europe Ltd.; 0.5 µl/100 µl) for 30 mins at 4 °C. Cells were washed twice again and then analyzed by flow cytometry.

## Flow cytometry

All flow cytometry data were acquired using either a FACSCalibur or FACSCanto II with data analysis performed using FlowJo (all from BD Biosciences). Gating strategies are shown in Supplementary Fig. 20.

## SAXS data collection and analysis

SAXS data were collected at the ESRF beamline BM29[39,40]. Samples were loaded using a SEC-SAXS set-up, passing through a SEC-3 column (300 Å pore size, 4.6 mm i.d., 300 mm length, Agilent) coupled to a Shimadzu HPLC system at a flow rate of 0.3, 0.25 or 0.2 mL/min using 50 mM HEPES, 150 mM KCl pH 7.5 as the SEC buffer, before entering a 1 mm diameter quartz glass capillary. Scattering images were collected using a Dectris Pilatus3 2 M with a sample-to-detector distance of 2.867 m. Measurements were recorded at 20 °C. See Supplementary Tables 13, 14.

BioXTAS RAW (version 2.3.0)[41] was used for data processing and primary data analysis, including buffer subtraction determination of the R$_g$ and D$_{max}$, and Kratky analysis. The SAXS data were deposited into the Small Angle Scattering Biological Data Bank (SASBDB)[42]. See Supplementary Tables 4–7.

## In silico design of antibody variants

Design of the new antibody variants followed two approaches; manual visual inspection and analysis using the program Disulfide by Design 2.0 (DbD2)[33]. In both cases, the structure of the anti-$h$CD40 ChiLob7/4 C232S κC214S F(ab')$_2$ was used as the starting structure (PDB:6TKE). The F(ab')$_2$ structure was generated by applying symmetry operators to the single F(ab') found in the asymmetric unit. Visual inspection was performed in PyMOL (The PyMOL Molecular Graphics System, Version 2.5.0, Schrödinger, Inc.), with the distance measurement and angle measurement wizard used for identifying residues amenable to forming disulfide bonds if mutated to cysteine. DbD2 was run using standard parameters, looking for potential inter-chain disulfide bonds with a χ$^3$ angle of − 87° or + 97° ± 30 and Cα-Cβ-Sγ angle of 114.6° ± 10°.

## Protein crystallization, data collection and data processing

Protein crystallization was performed by sitting drop vapor diffusion in 96 well 3-drop Intelliplates (SwissSci, Switzerland) using the Oryx8 protein crystallization robot (Douglas Instruments, UK). The TCR/pMHC optimized protein crystallization screen (TOPS)[43] and commercially-available MORPHEUS screen (Molecular Dimensions)[44] were used as entry screens at 21 °C. Proteins were set up in crystal screens at a concentration of 10 mg/mL in a buffer of 50 mM HEPES, 150 mM KCl pH 7.5. Initial crystal hits were used to prepare seed stocks with MicroSeed Beads (Molecular Dimensions). Microseeding experiments were performed using the Oryx8 protein crystallization robot (Douglas Instruments, UK) with the entry TOPS and MORPHEUS screens. Final crystallization conditions for the cross-over + T222C κE123C variant yielded needle-shaped crystals in the trigonal space group P321, grown from 0.1 M TRIS pH 8, 15 % PEG 4000, 0.2 M (NH$_4$)$_2$SO$_4$ (native data, ID30A-3[45], ESRF, Grenoble, France) and 0.1 M TRIS pH 8, 25 % PEG 4000, 15 % glycerol (SAD data collection, I23[46], Diamond Light Source, Oxford, UK). Crystals for the cross-over + K228C variant were similar in size in 3-dimensions and grew in the orthorhombic space group P2$_1$2$_1$2$_1$, grown from 0.1 M TRIS pH 7.5, 15 %

PEG 4000, 15 % glycerol (SAD data collection, I23, Diamond Light Source, Oxford, UK). Structure determination used the CCP4i2 graphical user interface of the CCP4 program suite[47,48]. Details on data collection, structure determination, and refinement can be found in the Supplementary Information.

## MD simulation

For ChiLob7/4 hIgG2 C232S + C233S, C233S κC214S and C232S κC214S (cross-over), the protocol for MD simulation of F(ab')₂ variants has previously been described[20]. For ChiLob7/4 hIgG2 C232S + K228C κC214S (cross-over + K228C) and C232S + T222C κE123C + κC214S (cross-over + T222C κE123C), model completion for the missing loops was performed using MODELER[32], extending the model to the pepsin cleavage site as determined by mass spectrometry (Supplementary Figs. 21, 22). Where multiple conformations had been fitted to the electron density, the models were pruned to the major conformation. Protonation was performed using the H++ server (standard settings, pH 7.5)[49] and PDB2PQR (using PROPKA to assign protonation states at pH 7.5 with the AMBER ff99 force field)[50]. Atomistic MD was performed using GROMACS (version 2022.4)[51]. Protonated structures containing crystal waters, were solvated in a truncated octahedron box with a 31 Å buffer between the protein and the edge of the box, filled with pre-equilibrated Simple Point Charge water molecules[52]. The system was neutralized through the addition of chloride ions, followed by the addition of Na+ and Cl- ions to achieve a final concentration of 150 mM NaCl solution. Protein atoms were represented by the Amber ff14sb force field[53] and ions were represented by Joung and Cheatham[54], with the TIP3P water model[52]. Further system setup details are shown in Supplementary Table 15.

Energy minimization was performed using 15,000 steps of the steepest descent protocol, with a maximum step size of 0.01 nm. The system was equilibrated first under the NVT ensemble, with heating to 300 K over 100 ps using the V-rescale (modified Berendsen) thermostat[55] with position restraints applied. Equilibration continued under the NPT ensemble at 1 bar for 100 ps using the Parrinello-Rahman barostat[56] with a time constant of 2 ps and position restraints applied. For each variant, three independent equilibration runs were performed, starting with different random number seeds for velocity generation. From these three sets of equilibration runs, three independent MD runs were performed using the leapfrog integrator with a 2 fs timestep. The Nose-Hoover thermostat[57] was used to maintain the temperature at 300 K, and the Parrinello-Rahman barostat was used to maintain the pressure at 1 bar. MD simulations used periodic boundary conditions and a 1.2 Å cut-off for short-range non-bonded interactions with a switching function at 1.1 Å. Particle mesh Ewald summation[58] was used for long-range electrostatics. An analytical dispersion correction was used to account for long-range van der Waals interactions. The LINCS algorithm[59] was used to constrain hydrogen bonds. Each independent MD run for each variant was performed for 2.1 μs with the first 100 ns discarded. Models were extracted from the resulting trajectories at 1 ns intervals yielding multiple structural states for each variant. $R_g$ as a function of time for the three repeats is shown in Supplementary Fig. 23.

## SAXS structure fits

$\chi^2$ fits between the crystal structures or the MD-generated models with the experimental SAXS data (cut to $q = 0.2$ Å$^{-1}$) were generated using CRYSOL 3.2.1[60] (from the ATSAS suite[61]) in batch mode with constant subtraction enabled, 50 spherical harmonics, explicit hydrogens and a water shell. Error-weighted residuals ($\Delta/\sigma$) were calculated using I(exp)-I(model)/ σ, and CorMap p analysis[62] was performed using PRIMUS (from the ATSAS suite[61]). For the variants C232S + C233S and C233S κC214S that were used as controls, ensemble fitting was performed as previously described (see Supplementary Methods and Supplementary Fig. 15).

## Statistics and reproducibility

Statistical analyses were performed in GraphPad Prism software (Graphpad), with details of statistical tests shown in figure legends. Reproducibility, including independent biological replicates, is shown in figure legends.

## Reporting summary

Further information on research design is available in the Nature Portfolio Reporting Summary linked to this article.

## Data availability

Crystallographic data has been deposited in the Protein Data Bank (PDB) with accession codes 8PUL and 8PUK (Supplementary Table 8). SAXS data has been deposited in the Small Angle Scattering Biological Data Bank (SASBDB) with accession codes SASDUB8, SASDUC8, SASDUD8, SASDUE8, SASDUF8, SASDUG8, SASDUH8, SASDUJ8, SASDUK8, SASDUL8, SASDSC7, SASDSD7, SASDWN2, SASDWP2, SASDWQ2, SASDWR2, SASDWS2, SASDWT2, SASDWU2, SASDWV2, SASDWW2, SASDWX2, SASDWY2 and SASDWZ2 (https://www.sasbdb.org/project/1827/). MD simulation input and output data are available in the Zenodo repository at doi.org/10.5281/zenodo.12582681. Raw SAXS data for BM29, proposal MX2373 is available at doi.org/10.15151/ESRF-ES-748850843 and doi.org/10.15151/ESRF-ES-771372332 and for BM29, MX2639 is available at doi.org/10.15151/ESRF-ES-1830158910 and doi.esrf.fr/10.15151/ESRF-ES-1893933559. Raw crystallography X-ray data is available at doi.org/10.15151/ESRF-ES-686689060 for ID30A-3, proposal MX2373, at https://ispyb.diamond.ac.uk/dc/visit/mx29835-1/id/8064344 for I23, proposal number MX29835-1 and at https://ispyb.diamond.ac.uk/dc/visit/mx29835-9/id/8379953 for I23, proposal number MX29835-9. Source data are provided with this paper as a Source Data file. All other data needed to evaluate the conclusions in the paper are present in the paper or the Supplementary Information. Request for materials will be subject to a standard MTA with the University of Southampton, due to potential commercial development. Materials can be accessed via request to Mark Cragg (msc@soton.ac.uk) after completing an MTA. Materials can be made available for academic use, and only for commercial use if commercial agreements are put in place. The expected timeframe for response to access requests is 4 weeks. Once access has been granted, materials will be made available in perpetuity. Source data are provided in this paper.

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

## Acknowledgements

Funding was provided by CRUK grants DRCPGM\100039, C1477/A20537, C34999/A18087, C328/A25139, DRCDDRPGM-Apr2020\100005 and BBSRC BB/Y009339/1. IRIDIS is funded by the University of Southampton. Time on the ARCHER2 High-Performance Computing Facilities was awarded through the UK High-End Computing Consortium for Biomolecular Simulation (HECBioSim) (EPSRC grant EP/R029407/1 and EP/X035603/1). We acknowledge the European Synchrotron Radiation Facility (ESRF) for the provision of synchrotron radiation facilities under proposal numbers MX2373 and MX2639, and we would like to thank Mark Tully and Stephanie Hutin for assistance and support in using beamline BM29 and Christoph Mueller-Dieckmann for assistance and support in using beamline ID30A-3. We would also like to thank Diamond Light Source for beamtime (proposals MX29835-1 and MX29835-9), and staff at beamline I23 for assistance with crystal harvesting and data collection. We thank Chris Holes at the Macromolecular Crystallization Facility, Biological Sciences, for support with crystallization experiments. We would like to thank The Membrane Protein Laboratory at the Diamond Light Source for access to the Prometheus NT.48 instrument, and Peter Harrison for support with the experiment. The Membrane Protein Laboratory is funded by a grant from the Wellcome Trust (223727/Z/21/Z) with additional support provided by Diamond Light Source and the Research Complex at Harwell, both Instruct-ERIC centers. The authors acknowledge the use of the University of Southampton's IRIDIS High-Performance Computing Facility and the ARCHER2 UK National Supercomputing Service, as well as associated support services, in the completion of this work.

## Author contributions

Conceptualization – H.F., I.G.E., J.W.E., M.S.C., and I.T.; Investigation – I.G.E., H.F., C.A.P., T.I., C.I.M., P.J.D., J.H., and K.T.J.M.; Resources – C.A.P., T.I., H.T.C.C., C.I.M., P.J.D., M.S.C., and I.T.; Methodology – H.F., I.G.E., and J.W.E.; Formal Analysis, Data Curation, and Validation – I.G.E., C.M.O., H.F., J.W.E., M.S.C., and IT; Visualization – I.G.E. and H.F.; Supervision – J.W.E., M.S.C., IT., and H.F.; Writing – Review and Editing – I.G.E., H.F., J.W.E., M.S.C., and I.T.; Writing – Original Draft – I.G.E. and M.S.C.; Funding Acquisition – J.W.E., M.S.C., and I.T.

## Competing interests

M.S.C. acts as a retained consultant for BioInvent International, consults for several other biotech companies, and receives institutional payments and royalties from antibody licenses. He has received research funding from BioInvent, GSK, iTeos, UCB, Surrozen and Roche. This work is related to patent Family WO 2015/145360 protecting antibodies containing modified $hIgG2$ domains which elicit agonist or antagonistic properties. JWE receives funding from: G.S.K., A.Z., Astex, UCB, dstl, Diamond Light Source, and exScientia. All other authors declare that they have no competing interests.
