## [Transparent Peer Review file · Nature Communications]

Structure-guided disulfide engineering restricts antibody conformation to elicit TNFR agonism

Corresponding Author: Dr Ivo Tews

Version 0:

Reviewer comments:

Reviewer #1

(Remarks to the Author)

Summary:

In this manuscript by Elliott and colleagues, the authors implemented a hinge disulfide engineering campaign to alter antibody conformation, resulting in modulation of agonistic activity by an antibodies against 2 members of the tumor necrosis factor receptor family, human 4-1BB and human CD40. They also use structural insights to predict additional constraining disulfide interactions that further augment agonistic activity of the resulting antibody. Overall, the topic is timely, as agonistic antibody therapeutics are of growing interest in the drug development space, the experimental approaches are rigorous, and the manuscript is well-written. A few points of revision are noted below. Upon their incorporation, we would recommend publication of this important work in Nature Communications.

Specific Points:

- 1) The acronym Fc R should be defined.
- 2) For Figure 4B, statistical comparisons at each point would be helpful. For instance it seems that activation induced by the cross-over+228C and cross-over+T222C kE123C variants saturate at a slightly higher level than the parental antibody. Is this significant?
- 3) Signal activation studies in primary B cells (via western blotting or flow cytometry analysis) would be helpful.
- 4) Quantification of Figure 4C would be beneficial. The cross-over+228C and cross-over+T222C kE123C variants look similar to hlgG2. Are there significant differences?
- 5) Urelumab and another anti-41BB antibody, utomilumab failed in the clinic. This should be discussed, and it would be helpful to provide a broader perspective on the field of immune agonists and how their approach might inform clinical design strategies.

Reviewer #2

(Remarks to the Author)

In the present manuscript, the authors extend the concept of superior antibody functionality through enhancing the rigidity of the molecule by engineering additional disulphide bonds in the hinge region of human IgG2 antibodies. In particular, they first demonstrate that the previously presented IgG2 formats, designed for graded flexibility using cysteine-to-serine exchanges in the hinge region, and tested in the context of anti-CD40 directed variable regions, are also most active agonists acting upon the immune checkpoint molecule 4-1BB, when the most rigid format of the hinge is chosen. SAXS studies for both antibodies confirm the same trends in the radius of gyration (Rg) and maximum intramolecular distance (Dmax) for both studied antibodies. Indeed, the superior agonism of least flexible mutants is demonstrated with the effect on the antigen-positive cell line, which expresses GFP upon activation of NFkappaB.

Further, two IgG2 variants with additional paired cysteines, one with an additional disulfide bond between the heavy chains and one with an additional disulphide bond between the opposing heavy and the light chain, are predicted in silico based on the parental C232S kappa-C214S format, and experimentally confirmed by crystal structures. Interestingly, the disulphide bond between C127 and C233 was also formed. Evidence of further restriction of conformational flexibility is delivered by experimental SAXS data and these are then used for ensemble optimisation of conformational pools generated by extensive

molecular dynamics simulations over 6 microseconds. These variants show potentiated agonistic activity as anti-CD40 antibodies in the model test cell line, and can elicit stronger adhesion and activation of primary human B cells isolated from PBMCs. Importantly, the binding affinity towards the cognate antigen is not strongly affected as shown with SPR and cell-binding experiments; the differences are however also dependent on variable regions and have to be determined on case-to-case basis.

Overall, this is a very well designed, methodologically diverse and elegantly performed study, and the manuscript is conclusively written and interesting to read. The outcomes are of high translational value and the novel format will certainly raise the interest of scientific community. In this view, I would ask the authors to consider including (in the supplementary material) the data on the expression level of the novel constructs comparing with IgG1 and IgG2 which they use as controls, and the initial SEC data after protein A purification as well as the monomer yield after SEC. Stability data would also support the utility of the concept.

Minor remarks, only to enhance the value of the manuscript:

Figure S3a, receptor binding of 4-1BB: isotype controls are missing

Please check the format of the reference 19 in the supplementary materials.

Reviewer #3

(Remarks to the Author)

Review of Elliot et al., Structure-guided disulfide engineering restricts antibody conformation and flexibility to elicit TNFR agonism.

A great deal of work has gone into this study, and I commend the authors for all of their efforts, and the study is interesting suffice to say. So well done.

Fundamentally, however, I find the manuscript is very confusing. It may have to do with the numbering of the amino acids, or the way the results are presented, or how the results are organized...but I am overall totally confused as to the flow of the manuscript and what it is being presented, when and why (or is not being presented and why).

So, from my estimation, there is a lot to do regarding three critical points.

Does the work support the conclusions and claims, or is additional evidence needed?

In general I think the evidence is there, but the formulation of the results is not clear.

Are there any flaws in the data analysis, interpretation and conclusions? Do these prohibit publication or require revision?

Yes - refer to my comments below.

Does the work meet the expected standards in your field?

...for the SAXS sections almost, except the errors are missing on key parameters - refer to my comments below.

Okay, here is an example:

Take Figure 2, panel A.

There is C127 linking to C233, and vice versa, C233 cross linking to C127. Then this construct is called 'C232S kC214S – 'cross-over'. So where is the serine mutation in this construct? Why in some points in the text does the kappa get removed from the C214S annotation (for example in Figure S8 and Sup Table S6)? Is there a difference, for example between C214S and kC214S? If so can this be easily explained or shown?

Then panel A is described as 'Starting model 6TKE'. When I download 6TKE from the PDB, then generate the dimer of dimers, I cannot find this cross over in the structure? Is there supposed to be a cross over? Or is this the mutant crystal structure.

Basically, when taking into account all 12 constructs, I don't know what I am looking at, and try as I may I cannot get a clear idea as to what construct is being referred to at any one time in the manuscript. I would recommend making schematic diagrams actually mapping *all* of the mutations and constructs listed in the manuscript, so the reader gets an idea what cysteine disulphide is forming in which construct and what cysteine has been mutated to serine...for *both* ChiLob7/4 anti-hCD40 and SAP 1.3. For example, I have no idea what disulphides are present in hIgG1 or hIgG2, or how the disulphide pattern is different between these two 'controls.'

Then, there are statements like:

"The global conformation observed in the crystals was consistent with previously characterised hIgG2 F(ab')₂ variants19 (PDB 6TKB, 6TKC, 6TKD, 6TKE, 6TKF)."

What is meant by 'The global conformation observed...' How did the authors quantify this observation? For example, if I take the dimer-of-dimers from 6TKE and then align to the dimer-of-dimers of 6TKB, there is significant rearrangement in the conformation. The 'global' Ca RMSD is 2.7 Angstroms between these two structures, and in addition, the Rg of 6TKB is 37.9 Angstrom compared to an Rg of 36.9 Angstrom for 6TKE – so there are measurable, and significant, differences here. Is

there a difference in the RMSD Ca or the Rg of the new crystal structure variants (8PUL and 8PUK) compared to the other Rg of all of the other variants (6TKB, 6TKC, 6TKD, 6TKE, 6TKF). Surely this important information to know and back up the statements about the new engineered mutants effects on the structure (in terms of 'global compaction')?

Question – don't you need to do peptide mass fingerprinting (mass spectrometry) to confirm the disulphide linkages that you think have formed have indeed formed for all of the constructs (for all 12 variants used in the project?)

Question – does AlphaFold3 support the disulphide crossover modelling and expected crosslinks, as engineered by the authors. This should be done – I believe AF3 can do this.

So now to the SAXS data.

1) I realize that there are a lot of constructs used in the manuscript, but I think it is necessary to systematically present the results in a clear fashion for all of the constructs (as in all 12) for comparative purposes. Again I find it very difficult to follow what is being talked about in the main text, or in the supplementary information. For example, in the main body of the text, the authors primarily focus on ChiLob7/4 anti-hCD40 variants for the SAXS analysis and modelling...is this correct? What happened to the SAP 1.3 analysis? Did I miss the modelling and results for SAP 1.3 – aside from the tables the results, and a passing mention in Figure 1, the analysis of SAP 1.3 seem to have gone by the wayside? If the aim is to demonstrate changes in conformation induced by disulphide engineering, then surely the alternative SAP 1.3 need to be analysed in the same way as ChiLob7/4 anti-hCD40? I go to the supplementary information and also cannot find anything on SAP 1.3, except in the SAXS reporting tables? For example, Supp fig 8- I don't know what base construct this is from...I assume ChiLob7/4 anti-hCD40? What happened to the equivalent analyses for SAP 1.3 (EOM modelling, etc)? This must be included and in detail.

2) How is Figure S7 different from Figure 1c, aside from the fact that on Figure S7 the C232S + C233S dimensionless Kratky plot is missing?

3) Figure 1 and Figure S7 it is stated that:
"Graphs show dimensionless real-space Kratky"

...real-space? Simply delete 'real-space' in the description . Dimensionless Kratky plots are just that, dimensionless!

4) Place error bars on the SAXS data, and dimensionless Kratky plots.

5) It is necessary to quote the errors on Rg in both the text and Tables (e.g., Table 1, Supp Table 2, etc)!

6) It is stated that:
"Error-weighted residuals (Δ/σ) were calculated using $I(\text{exp})-I(\text{model})/\sigma$, and are displayed as lines after fast Fourier transform smoothing."

What is fast Fourier transform smoothing and why would you need to do this if the smoothed fit to the data is already provided from EOM? You'd simply calculate the error-normalized residual plot using this smoothed fit.

7) For the $p(r)$ analysis, although the authors quote Rg and Dmax values, the issue is:

i) I cannot find any plots of the $p(r)$ comparing the constructs. A comparative plot should be included, where the $p(r)$ are all scaled for direct comparison, for example scale the data to a common point (e.g., a scaled overlay), or normalize $I(0)$ to 1 and calculate $p(r)$, etc.

ii) I cannot find how the authors assessed the fit to the data of the calculated $p(r)$ in reciprocal space i.e., reciprocal-space fits of $p(r)$ to the data. You understand the issue here, if the $p(r)$ do not fit the experimental data, then comparing Rg and Dmax values has little meaning! So, reciprocal space fits to the data and quality of the fits needs to be included, and reported in the Tables.

8) It is stated that:

"Together, our observations confirm that our disulfide- engineered variants evoke a more restricted degree of conformational flexibility, as well as greater compactness, compared to the parental C232S kC214S variant."

Sorry, I do not agree with this based on the data presented in the manuscript (or alternatively caused by my confusion about what construct is being referred to).

When I compare the following three SAXS profiles directly for the ChiLob7/4 variant, that have all been scaled together relative to point 20 of each data file:

SASDSC7 hIgG2 C232S+T222C kE123C+C214S
SASDSD7 hIgG2 C232S+ K228C kC214S
SASDUL8 hIgG2 C232S kC214S

After scaling and comparing, these profiles are statistically identical. Do the authors not find this interesting considering all

three contain hlgG2 C232S kC214S...(although this latter mutation is also listed as C214S and not kC214S, mutation in Table S6. Is there a difference between kC214S and C214S?) In addition SASDUH8 (hlgG2) is also very similar to the three entries listed above when you scale the data to the 20th point or so (could be incorrect background subtraction)...

If SASDSC7, SASDSD7, SASDUL8 and potentially SASDUH8 (after correct background subtraction) are indeed identical then how does this direct observation influence the statement:

"Primary analysis of the SAXS data indicated that both designed variants were slightly more compact than the parental C232S kC214S in solution, with decreased Rg and Dmax values in the distance distribution function."

This reviewer fundamentally does not agree. Within error all three(four) profiles and the subsequent structural parameters extracted from the SAXS data – especially the Rg – are effectively the same, indeed all engineered mutant construct (hlgG2 C232S+T222C kE123C+C214S, hlgG2 C232S+ K228C kC214S and hlgG2 C232S kC214S) are similar or the same as SASDUH8 (hlgG2) within measurement error. The only differences I can see in the ChiLob7/4 cohort are:

SASDUG8 hlgG1 – is unique
SASDUJ8 hlgG2 C232S+ C233S – is unique
SASDUK8 hlgG2 C233S kC214S – is unique

Then SASDSC7, SASDSD7, SASDUL8 and probably SASDUH8 can probably be grouped together as the same (effectively).

Note 1) One cannot resort to Dmax as an estimator of difference, unless Dmax is hugely significantly different. Please remember that Dmax is just a number to solve an integral. It is not directly measured. Consequently it has a huge error, especially if the data do not transform well at high values of r – probably +/-5 angstroms at best. This impacts e.g., the presentation of the plot in Figure 3 e.

Note 2) To really assess whether the changes in Rg noted in the text are statistically different, the authors need to go back and calculate the Rg correlation through the SEC elution peak, per subtracted frame, to obtain an idea as to the variance on Rg through the SEC elution (for all of their constructs). Only then can statements be made about differences in Rg taking into account the errors on Rg in the finally processed data.

For the other cohort of SAXS data – SAP 1.3 - that needs significant expansion and presentation in the manuscript (if it has been published before, simply referring to previous publications is insufficient here. The results need to be presented, irrespective).

Although not statistically identical, SASDUC8, SASDUB8 and SASDUE8 are very similar (again scaling from point 20), where SASDUE8 is just hlgG2. Again, how do the additional disulphide mutations significantly affect the structure of those constructs engineered with additional mutations?

Then the authors need to do a comparative analysis of SAP 1.3 variants with ChiLob7/4 – are the SAP 1.3 variants more or less 'flexible' than the ChiLob7/4 variants? The point of this is to demonstrate that it is the disulphides and NOT some other effect, e.g., repositioning of surface residues induced by the disulphides, that are causing the change in conformation, or structural sampling. The authors have mentioned torsional angles, etc in the manuscript, but does this hold true for both SAP 1.3 and ChiLob7/4?

9) The MD and EOM section.

It is the opinion of this reviewer that the MD simulations need to be revisited in order to correctly interpret the conclusions drawn in the manuscript. The major issue – and aside from C232S+C233S – is that the MD simulations are *not* sampling sufficient structures anywhere near the Rg determined from the experimental SAXS data. It is interesting the initial pool of MD simulated structures are significantly more compact in terms of frequencies relative to the experimental data. For example when I look at Figure 3 of the main text, the most frequent Rg is about 37.5 Angstroms, well below the experimental Rg of either construct (38.6 and 38.7). So, EOM is simply being forced to select a minimized ensemble that chooses less frequent models with higher Rg to fit the data (as the results show). The question is, would it not be better to start the MD simulation on a model that starts with an Rg similar to the Rg of the experiment – generate an ensemble from this, and then assess the sampling? This 'forcing EOM to select limited models from restricted under-sampled pools of starting structures' invalidates the comparison of Rflex between the constructs. Considering that SASDSC7, SASDSD7, SASDUL8 are identical SAXS profiles (after scaling) – and SASDUH8 is pretty close to these, in my mind there is a good argument to be made that these data could be described as *single* models. Did the authors try and simply fit single models to the data? For example in the main text of Figure 3, I think am looking at a comparison of

SASDSC7 hlgG2 C232S+T222C kE123C+C214S
SASDSD7 hlgG2 C232S+ K228C kC214S

...which are statistically equivalent profiles (after scaling), but the Rflex bar graph suggests differences here, where there are none because the SAXS data are the same!

As an example of single model generation – if I take the SASBDB models deposited for SASDSC7 and SASDSD7 and

simply do a CRY SOL 2.8.3 fit to the data, out to an q_{\max} of 0.4 inverse Angstroms:

```
crystal *.pdb *.dat -lm=50 -ns=256 -sm=0.2 -un=1 -cst -err
```

For SASDSD7_fit2_model4, the χ^2 of the model fit to the data is pretty good, at 1.21, while for the other entry, SASDSC7_fit2_model1 the χ^2 is 1.24, again not that bad.

Using FoXS for the model fitting routine also provides a similar result for the single model representation fit to the data, using the maximum q -value (SASDSC7_fit2_model1, $\chi^2 = 1.26$; SASDSD7_fit2_model4, $\chi^2 = 1.3$).

...so these single model representations are not that bad, and can probably be refined even further. Why not just calculate the fits to the data of the individual models generated from MD, with a focus on those with an R_g close to that of the SAXS experiment? If you can indeed develop single models that fit the data better than the EOM models, where does this put the conclusions reached about conformational-flexibility? This needs to be investigated.

10) R_g and flexibility.

As you know SAXS data are ambiguous because of the spherically averaged scattering amplitudes. Therefore it is necessary to at least provide a statement to the reader that having a shorter R_g does not necessarily correlate with a decrease in flexibility, as implied in the manuscript, or vice-versa, having a larger R_g correlates with an increase in flexibility. There are clearly instances where modular proteins can have the same molecular weight, but be less flexible, and more extended, giving a larger R_g ; and conversely, it is conceivable that something that is on average more mass-centred has multiple mass-centred conformations (smaller R_g but still flexible). In effect, different conformations can have the same R_g , so R_g in and of itself cannot be used to assess flexibility. How ambiguous are the data (e.g., using AMBIMETER to assess this)?

Final comments:

i) If the activity binding assays etc were done on full mAb variants, the authors should also perform the same types of assays using the truncated variants used for SAXS? The Fc does sometimes affect the conformational sampling of the Fab?

ii) Maybe short statement to include that the production of the mAb do not incorporate glycans?

iii) To me there is far too much speculation in the discussion, specifically statements that have no experimental support provided in the main text of the manuscript. For example:

“The mechanism by which conformational restriction impacts on biological activity is likely mediated by the ability of the different mAb variants to promote receptor clustering... Disulfide-engineered variants with a more compact molecular arrangement and restricted conformational flexibility may bind receptors in closer proximity, promoting efficient clustering of receptors in the cell membrane and triggering activation of downstream intracellular signalling pathways, leading to cellular activation. In contrast, mAb variants with greater conformational freedom, which sample very extended conformations, may engage two distant receptors. This would be less likely to stabilise receptor clusters and thus be incapable of surpassing the receptor signalling threshold to promote strong agonism.”

In summary, I think to be more convincing requires a reworking of the manuscript in terms of organisation and flow, along with detailed analyses of *all* of the protein variants used, which included a comparative analysis of the SAXS results.

Reviewer #4

(Remarks to the Author)

In this manuscript, the authors show how disulphide engineering can enhance the agonist activity of antibodies. The hypothesis is that the effect is due to restricted flexibility in the antibody. The authors probe the flexibility using SAXS and atomistic Molecular Dynamics simulations. Due to the rather small changes in the size of the antibodies, they work with Fab₂ fragments. This aspect of the manuscript raises significant questions that the authors will need to satisfactorily address:

1. Is the behaviour of the solvated fragments representative of that in the full antibody, where the Fc will sterically influence the relative Fab positions and their dynamics? This aspect needs more discussion.

2. In the simulation work, the authors report “reweighted ensembles” without proper explanation of what this means and why it is justified. On the face of it, it seems like they select simulated conformations that align with experimental data, and discard the rest, so should we be surprised when the end results agree?

3. The changes to the size of the antibody fragments are small, yet data is presented without statistical analysis (not even errorbars in Table 1). This must be properly addressed.

4. The hypothesis that the more rigid constructs promote receptor clustering could be complemented by simulation data for the relative CDR conformations. It seems unlikely to me that reduced flexibility would enhance receptor interaction, however

some further insight might be possible to support the picture.

Consequently, I do not recommend accepting the manuscript in its current form.

Version 1:

Reviewer comments:

Reviewer #1

(Remarks to the Author)

All my comments have been addressed and I endorse publication of this important work.

Reviewer #2

(Remarks to the Author)

In the revised version of the manuscript, the authors have added the data on their novel antibody constructs demonstrating their monomeric behavior in SEC-HPLC in native conditions, and their high thermal stability measured with DSF, both of which supports the high developability potential of their proposed format and is relevant to increase the importance of this study with imminent high translational value. Further, they have improved on concept presentation (Figure 1a), and reworded the findings for more semantical clarity. The study is an important contribution to the knowledge base on engineering of antibodies for modulation of biological effects (in their case agonism) by altering the mobility of the arms rather than altering the binding sequence, which is of interest for several targets with validated clinical relevance (e.g. most in immunoncology).

Will the work be of significance to the field and related fields?

Yes

Does the work support the conclusions and claims, or is additional evidence needed?

Yes, well supported

Are there any flaws in the data analysis, interpretation and conclusions? Do these prohibit publication or require revision?

No

Is the methodology sound? Does the work meet the expected standards in your field?

Yes

Is there enough detail provided in the methods for the work to be reproduced?

Yes

Minor: line 92, something with the reference inline citations

Reviewer #3

(Remarks to the Author)

....and that's how you do it.

I do not know who is responsible for reprocessing and analysing the SAXS data, modelling, interpretation and presentation, but they should be commended. Basically a transformative text when compared to the first version of the manuscript, so well done.

i) I only have very minor comments, all are in relation to just double checking consistency across text, tables and for the SAXS the parameters reported in SASBDB. Just go through all of the numbers and parameters (a tricky task for sure), e.g., the R_g quoted in the tables, text, figures, Figure panels and SASBDB etc, to make sure all are consistent for each construct.

ii) Tiny things like, in the Methods: 'SAXS structure and fits' it is quoted:

"...experimental SAXS data (cut to $q = 0.2$)"

--the unit on q is missing.

Another example: In the Supp Info, Figure S12, at the end of the legend it refers the reader to Figure S12.

All of this can be done with careful re-reading, and input from the Senior editor.

iii) Only one point I would like to see in the text: A short statement somewhere about glycosylation. I brought up the topic of glycans in my review because I have a major bug about glycosylation in general. People often omit the glycosylation state(s) in their SAXS models, because they think that glycans miraculously do not scatter X-rays (which is of course complete nonsense). Of course, and especially for anything antibody-related, glycosylation could affect both stability and conformational sampling. Therefore, it would be good to include a statement somewhere in the text along the lines of how you responded to me, e.g.:

"The antibodies are produced in mammalian CHO cells, therefore mAb production results in Fc glycosylation. This occurs in all antibodies (including natural antibodies produced in vivo) at the N297 residue in the Fc domain. This glycosylation was not removed prior to our

14 biological assays or new IgG SAXS data collection. Lob7/4 has no glycosylation sites in its F(ab') regions so the Lob7/4 F(ab')₂ fragments assessed by SAXS do not have glycosylation, whereas SAP1.3 does have a glycosylation site in its F(ab') region."

iv) Okay, I let you get away with the statement:

The good agreement of individual models extracted from the MD-generated conformational pools indicates that the engineered variants, as well as the parent, are particularly rigid, as the SAXS data can be described well with a single structure.

---not technically true in and of itself (ask your SAXS expert why). You might want to re-word?

I still think that - and although greatly de-emphasised relative to the first version of the manuscript - the statement in the discussion:

"The restricted conformational diversity of disulfide-engineered variants may retain receptors in closer proximity, promoting efficient clustering of receptors in the cell membrane, trigger activation of downstream intracellular signalling pathways, thus leading to cellular activation. In contrast, mAb variants with greater conformational freedom would be less likely to stabilise receptor clusters and thus be incapable of surpassing the receptor signalling threshold to promote strong agonism."

...is a bit dangerous to conclude based on the data presented *this* study....You really are looking at having to do some sort of reconstituted liposome/receptor + fluorescence microscopy work here to back up this statement. Receptor clustering is assumed, but there is no data in the manuscript to support the clustering hypothesis. I of course do not exclude this hypothesis, but it would not shock me if another process within or at the surface of the cell could be involved. And then there is the stability of the antibodies to consider, not in terms of thermodynamic stability, but something like resistance to secondary proteolysis/processing events, redox-state(s), blah blah blah. It all depends on how the antibodies engage with the receptors correct? Where they bind and in what stoichiometry(s) they bind, what effect they have on receptor conformation, and maybe even recruitment of secondary/tertiary components in/along the cell membrane (that might not be the actual targeted receptor) - could be a combination of all of this together? Basically it is complicated. I leave this over to the editor to decide.

In conclusion - check consistency, tidy up small stuff here and there. Should be fine.

Reviewer #4

(Remarks to the Author)

The authors have responded positively to my objections. I have no further qualms about proceeding to publication with this paper, other to note that the authors still state some results without corresponding +/- SD (e.g. page 7 hinge angles, Rg values).

Point by Point response to the Reviewers' Comments:

Reviewer #1

Summary:

In this manuscript by Elliott and colleagues, the authors implemented a hinge disulfide engineering campaign to alter antibody conformation, resulting in modulation of agonistic activity by an antibodies against 2 members of the tumor necrosis factor receptor family, human 4-1BB and human CD40. They also use structural insights to predict additional constraining disulfide interactions that further augment agonistic activity of the resulting antibody. Overall, the topic is timely, as agonistic antibody therapeutics are of growing interest in the drug development space, the experimental approaches are rigorous, and the manuscript is well-written. A few points of revision are noted below. Upon their incorporation, we would recommend publication of this important work in Nature Communications.

We thank the reviewer for these comments and their positive assessment.

Specific Points:

1) The acronym Fc γ R should be defined.

We have defined the acronym Fc γ R as Fc gamma receptor (introduction, line 58 of the original submitted manuscript).

2) For Figure 4B, statistical comparisons at each point would be helpful. For instance it seems that activation induced by the cross-over+228C and cross-over+T222C kE123C variants saturate at a slightly higher level than the parental antibody. Is this significant?

We thank the reviewer for this helpful suggestion. To compare the parental antibody (C232S κ C214S, cross-over) with each of the new engineered variants (cross-over + K228C and cross-over + T222C κ E123C), we performed a one-way ANOVA with Tukey's multiple comparisons test at each concentration of antibody in the NF- κ B GFP reporter assay. The data are included (Table S10 of the resubmission) and referenced in the results and figure legend. The analysis supports the presentation of data in the results section, concluding significantly higher levels of NF- κ B activation are observed at low concentrations with the engineered variants (cross-over + K228C and cross-over + T222C κ E123C) compared to the parental C232S κ C214S cross-over. The analysis also reveals significantly higher activity at the top concentration.

3) Signal activation studies in primary B cells (via western blotting or flow cytometry analysis) would be helpful.

We show activation of primary B cells (Figure 4d-f of the resubmission) as expression of the B cell activation markers HLA-DR, CD23 and CD86 measured using flow cytometry. The text of the figure legend has been updated to reference the method.

4) Quantification of Figure 4C would be beneficial. The cross-over+228C and cross-over+T222C kE123C variants look similar to hIgG2. Are there significant differences?

We used alternative methods to provide quantification of the agonism induced, reporting activation (Figures 4d-f of the resubmission) and proliferation (Figure 4g of the resubmission) of primary B cells. We have not found an image analysis tool that accurately and quantifiably resolves differences in the images but would prefer to keep Figure 4c as it gives an easy-to-grasp visual impression.

5) Urelumab and another anti-41BB antibody, utomilumab failed in the clinic. This should be discussed, and it would be helpful to provide a broader perspective on the field of immune agonists and how their approach might inform clinical design strategies.

We thank the reviewer for highlighting this point. We have added new text into the introduction discussing these points alongside a recent review from a leader in this field (ref 14) (after line 55 of the original submitted manuscript):

Several of these antibodies have reached clinical trials, including two anti-4-1BB mAb, Urelumab and Utomilumab¹⁰. However, these mAb eventually failed in the clinic due to problems with efficacy and toxicity, respectively¹⁴.

14 Claus, C., Ferrara-Koller, C. & Klein, C. The emerging landscape of novel 4-1BB (CD137) agonistic drugs for cancer immunotherapy. *mAbs* **15**, 2167189 (2023). <https://doi.org/10.1080/19420862.2023.2167189>

Reviewer #2 (Remarks to the Author):

In the present manuscript, the authors extend the concept of superior antibody functionality through enhancing the rigidity of the molecule by engineering additional disulphide bonds in the hinge region of human IgG2 antibodies. In particular, they first demonstrate that the previously presented IgG2 formats, designed for graded flexibility using cysteine-to-serine exchanges in the hinge region, and tested in the context of anti-CD40 directed variable regions, are also most active agonists acting upon the immune checkpoint molecule 4-1BB, when the most rigid format of the hinge is chosen. SAXS studies for both antibodies confirm the same trends in the radius of gyration (Rg) and maximum intramolecular distance (Dmax) for both studied antibodies. Indeed, the superior agonism of least flexible mutants is demonstrated with the effect on the antigen-positive cell line, which expresses GFP upon activation of NFkappaB.

Further, two IgG2 variants with additional paired cysteines, one with an additional disulfide bond between the heavy chains and one with an additional disulphide bond between the opposing heavy and the light chain, are predicted *in silico* based on the parental C232S kappa-C214S format, and experimentally confirmed by crystal structures. Interestingly, the disulphide bond between C127 and C233 was also formed. Evidence of further restriction of conformational flexibility is delivered by experimental SAXS data and these are then used for ensemble optimisation of conformational pools generated by extensive molecular dynamics simulations over 6 microseconds. These variants show potentiated agonistic activity as anti-CD40 antibodies in the model test cell line, and can elicit stronger adhesion and activation of primary human B cells isolated from PBMCs. Importantly, the binding affinity towards the cognate antigen is not strongly affected as shown with SPR and cell-binding experiments; the differences are however also dependent on variable regions and have to be determined on case-to-case basis.

Overall, this is a very well designed, methodologically diverse and elegantly performed study, and the manuscript is conclusively written and interesting to read. The outcomes are of high translational value and the novel format will certainly raise the interest of scientific community.

We thank the reviewer for these comments and their positive assessment.

In this view, I would ask the authors to consider including (in the supplementary material) the data on the expression level of the novel constructs comparing with IgG1 and IgG2 which they use as controls, and the initial SEC data after protein A purification as well as the monomer yield after SEC.

We have included these data (Tables S1 and Figures S1 of the resubmission).

The new Table S1 shows expression data for all constructs, allowing for comparison with the parental *h*IgG1 and *h*IgG2 controls, though one must caution to directly compare the antibody titre where they were produced using different methods (stably expressed CHO cell lines vs. transiently expressed CHO cells). We have also reported the final IgG purity in this table. SEC was only performed after protein A purification if there was >1% aggregation as stated in Methods.

The new Figure S1 shows HPLC traces for each of the constructs – these correspond to the final IgG purity values reported in Table S1. We have also added the following text to the Methods (line 457 of the original submitted manuscript):

Antibody titre and final IgG purity are shown in Supplementary Table S1, and HPLC traces are shown in Supplementary Fig. S1.

Stability data would also support the utility of the concept.

We have included these data (Table S2 and Figure S3 of the resubmission).

New Table S2 and Figure S3 now provide stability data that we have collected in response to the reviewer's comments. We used nano Differential Scanning Fluorimetry (nanoDSF) data for the ChiLob7/4 hIgG2 variants. The data indicate that all hIgG2 variants have similar thermal stability to the parent molecule. We have referenced this in the section that described the protein production (line 223 of the original submitted manuscript):

The new mAb variants were produced and characterised by SDS-PAGE and CE-SDS (Supplementary Fig. S12) and retained an equivalent thermal stability profile to the parental C232S κC214S cross-over variant (Supplementary Fig. S3, Table S2).

Minor remarks, only to enhance the value of the manuscript:
Figure S3a, receptor binding of 4-1BB: isotype controls are missing

We thank the reviewer for highlighting this and have added a hIgG2 isotype controls to this graph (originally Figure S3a, now Figure S5a).

Please check the format of the reference 19 in the supplementary materials.

Thank you.

Reviewer #3 (Remarks to the Author):

Review of Elliot et al., Structure-guided disulfide engineering restricts antibody conformation and flexibility to elicit TNFR agonism.

A great deal of work has gone into this study, and I commend the authors for all of their efforts, and the study is interesting suffice to say. So well done.

We thank the reviewer for these comments.

Fundamentally, however, I find the manuscript is very confusing. It may have to do with the numbering of the amino acids, or the way the results are presented, or how the results are organized...but I am overall totally confused as to the flow of the manuscript and what it is being presented, when and why (or is not being presented and why).

While reviewer 1 stated “the manuscript is well-written” and reviewer 2 attested “the manuscript is conclusively written and interesting to read” we agree that the nomenclature could be construed as confusing, and the accessibility could be improved. We have addressed this in two ways:

1. We include a new overview figure that serves as a reference for the nomenclature.
2. We used a consistent nomenclature and edited the m/s throughout to support the reader in following the numbering of amino acids.

So, from my estimation, there is a lot to do regarding three critical points.

Does the work support the conclusions and claims, or is additional evidence needed?
In general I think the evidence is there, but the formulation of the results is not clear.

We thank the reviewer for querying the conclusions based on the presented evidence and have addressed this carefully. A complete re-evaluation of the SAXS analysis was performed, following guidance by the reviewer, and new data SAXS data were collected on full IgGs, now included, also in response to comments made by reviewer 4.

A significant change in the interpretation of the data is removal of the multi-model analysis. Consequently, we removed statements on dynamics and flexibility from the description. The text has been revised and focusses on the concept of rigidifying the structure. The revised title reflects this change.

Are there any flaws in the data analysis, interpretation and conclusions? Do these prohibit publication or require revision?

Yes - refer to my comments below.

We have addressed the points raised by the reviewer, as detailed below.

Does the work meet the expected standards in your field?

...for the SAXS sections almost, except the errors are missing on key parameters - refer to my comments below.

Statistical analysis was performed and is now reported throughout, with error estimations included in the presentation of the data.

Okay, here is an example:

Take Figure 2, panel A.

There is C127 linking to C233, and vice versa, C233 cross linking to C127. Then this construct is called 'C232S κC214S – 'cross-over''. So where is the serine mutation in this construct?

We have revised Figure 1a and included new diagrams that show the topology of the disulfides. We indeed use the CxxxS nomenclature to reference cysteine/serine exchange variants. The resulting structure and disulfide topology is, however, from experiment and should thus not be used as naming convention. The nomenclature is consistent with our earlier publication¹.

In the example 'C232S κC214S' the nomenclature indicates that C232 is mutated to serine, and the κC214 is mutated to serine (κ stands for light chain, see below). We define this variant as "cross-over" of the main text (line 192 of the original submitted manuscript):

The C232S κC214S variant is henceforth referred to as cross-over.

Why in some points in the text does the kappa get removed from the C214S annotation (for example in Figure S8 and Sup Table S6)? Is there a difference, for example between C214S and κC214S? If so can this be easily explained or shown?

Thank you for spotting this. The Greek κ indicates the amino acid is found in the kappa light chain, so the κ should not be removed from the annotation. We have checked the text and figures to ensure consistent nomenclature.

Then panel A is described as 'Starting model 6TKE'. When I download 6TKE from the PDB, then generate the dimer of dimers, I cannot find this cross over in the structure? Is there supposed to be a cross over? Or is this the mutant crystal structure.

We thank the reviewer for the opportunity to clarify this. As there is only one F(ab') arm (heavy and light chain) in the asymmetric unit, we understand the reviewer has created the dimer of dimers by symmetry operation. It may be that the software used does not display the disulfide, but the distance between the gamma sulfur atoms of Cys225 and κC136 in the symmetry generated molecule is 2.7 Å, indicating a disulfide. Presence of the disulfide cross-over was experimentally determined using an anomalous scattering approach. Note that Kabat residue numbering is used, please refer to methods section (line 444 of the original submission). We clarified the nature of the disulfide across symmetry mates by adding a sentence (line 519 of the original submitted manuscript):

The F(ab')₂ structure was generated by applying symmetry operators to the single F(ab') found in the asymmetric unit.

Basically, when taking into account all 12 constructs, I don't know what I am looking at, and try as I may I cannot get a clear idea as to what construct is being referred to at any one time in the manuscript. I would recommend making schematic diagrams actually mapping *all* of the mutations and constructs listed in the manuscript, so the reader gets an idea what cysteine disulphide is forming in which construct and what cysteine has been mutated to serine...for *both* ChiLob7/4 anti-hCD40 and SAP 1.3.

For example, I have no idea what disulphides are present in hlgG1 or hlgG2, or how the disulphide pattern is different between these two ‘controls.’

This issue is now addressed in the revised Figure 1a that gives the schematic showing the topology of the disulfides described, complementing information given in Figure 2a. The constant regions of ChiLob7/4 and SAP1.3, and thus disulfide patterns, are identical.

Then, there are statements like:

“The global conformation observed in the crystals was consistent with previously characterised hlgG2 F(ab’)2 variants¹⁹ (PDB 6TKB, 6TKC, 6TKD, 6TKE, 6TKF).”

What is meant by ‘The global conformation observed...’ How did the authors quantify this observation? For example, if I take the dimer-of-dimers from 6TKE and then align to the dimer-of-dimers of 6TKB, there is significant rearrangement in the conformation. The ‘global’ Ca RMSD is 2.7 Angstroms between these two structures, and in addition, the Rg of 6TKB is 37.9 Angstrom compared to an Rg of 36.9 Angstrom for 6TKE – so there are measurable, and significant, differences here. Is there a difference in the RMSD Ca or the Rg of the new crystal structure variants (8PUL and 8PUK) compared to the other Rg of all of the other variants (6TKB, 6TKC, 6TKD, 6TKE, 6TKF). Surely this important information to know and back up the statements about the new engineered mutants effects on the structure (in terms of ‘global compaction’)?

In response to the reviewers’ query, we calculated R_g using CRY SOL (ATSAS 3.2.1) and calculated the RMSD to 6TKE for all structures using PyMOL (version 3.0.0). The values shown in the table below indicate larger differences observed for 6TKB and 8PUL. These are easily understood by crystal packing differences. The SAXS and MD analyses provided in the m/s avoid crystal packing artefacts.

PDB ID	Mutations	Variant name	R _g (Å)	RMSD to 6TKE (Å)	Space group
6TKB	C232S		37.41	1.859	P212121
6TKC	C233S		36.90	0.324	P321
6TKD	C239S		36.73	0.258	P321
6TKE	C232S κC214S		36.65	0	P321
6TKF	C233S κC214S	cross-over	36.95	0.586	P321
8PUL	C232S+K228C κC214S	cross-over + K228C	37.82	1.806	P212121
8PUK	C232S+T222C κE123C+κC214S	cross-over + T222C κE123C	36.59	0.870	P321

We appreciate some of the earlier wording was confusing and have removed the statement on global conformation.

Question – don’t you need to do peptide mass fingerprinting (mass spectrometry) to confirm the disulphide linkages that you think have formed have indeed formed for all of the constructs (for all 12 variants used in the project?)

A mass spectroscopic analysis would give the connectivity, but not the topology of disulfide connections across the dimeric F(ab')₂. We therefore use an anomalous scattering approach that unambiguously identifies sulfur positions, and from this, resolves the disulfide structure.

While this was performed for all ChiLob7/4 variants (in this and our previous manuscript¹) this was not possible for SAP1.3 as we were not successful in crystallising this antibody. However, the evidence from SDS-PAGE in the supplementary information (Figure S1 of the original submission, now Figure S2) suggests no differences in disulfide formation between ChiLob7/4 and SAP1.3.

Question – does Alphafold3 support the disulphide crossover modelling and expected crosslinks, as engineered by the authors. This should be done – I believe AF3 can do this.

AF3 may be able to predict this but experimental evidence such as presented in our manuscript should be the accepted and preferred way of the model generation on which we base our scientific interpretations.

So now to the SAXS data.

1) I realize that there are a lot of constructs used in the manuscript, but I think it is necessary to systematically present the results in a clear fashion for all of the constructs (as in all 12) for comparative purposes. Again I find it very difficult to follow what is being talked about in the main text, or in the supplementary information.

We believe this is now addressed with the revised Figure 1a, see comments above.

For example, in the main body of the text, the authors primarily focus on ChiLob7/4 anti-hCD40 variants for the SAXS analysis and modelling...is this correct? What happened to the SAP 1.3 analysis? Did I miss the modelling and results for SAP 1.3 – aside from the tables the results, and a passing mention in Figure 1, the analysis of SAP 1.3 seem to have gone by the wayside? If the aim is to demonstrate changes in conformation induced by disulphide engineering, then surely the alternative SAP 1.3 need to be analysed in the same way as ChiLob7/4 anti-hCD40? I go to the supplementary information and also cannot find anything on SAP 1.3, except in the SAXS reporting tables? For example, Supp fig 8- I don't know what base construct this is from...I assume ChiLob7/4 anti-hCD40? What happened to the equivalent analyses for SAP 1.3 (EOM modelling, etc)? This must be included and in detail.

In this manuscript, we tested the concept of cysteine variation that was previously established for ChiLob7/4 (as published¹) on another antibody, SAP1.3, targeting a different receptor, 4-1BB. We present dimensionless Kratky plots in Figure 1c and now include the full SAXS analysis in Table S4 and Figure S6 for ChiLob7/4 and in Table S5 and Figure S7 for SAP1.3.

For the study, to further increase agonistic activity we also start with ChiLob7/4. The reason to focus the analysis on this mAb is availability of experimentally determined crystal structures, as described from line 188 in the original submitted manuscript. The focus on ChiLob7/4 supports use of a structure-guided approach to introduce new disulfides.

We thank the reviewer for spotting the omission in Supplementary Figure S8 (now Figure S15) and have updated the legend.

2) How is Figure S7 different from Figure 1c, aside from the fact that on Figure S7 the C232S + C233S dimensionless Kratky plot is missing?

Figure 1C compares the dimensionless Kratky plots for Lob7/4 C232S+C233S, C233S κC214S and C232S κC214S with the *hlgG1* and *hlgG2* wildtype antibodies. Figure S7 compares the ChiLob7/4 new engineered variants (crossover + K228C and crossover + T222C κE123C) with the parental cross-over antibody from which they were designed (C232S κC214S) with the *hlgG1* and *hlgG2* wildtype antibodies.

3) Figure 1 and Figure S7 it is stated that:
“Graphs show dimensionless real-space Kratky”

...real-space? Simply delete ‘real-space’ in the description . Dimensionless Kratky plots are just that, dimensionless!

Dimensionless Kratky plots can be generated using the Guinier estimation of R_g or the real-space R_g obtained using the $P(r)$ function. While we used the latter in the original submission, with this submission we report the former, and consequently deleted ‘real-space’ from the text.

4) Place error bars on the SAXS data, and dimensionless Kratky plots.

The SAXS data was originally processed and analysed using ScAtter. For submission of the revision, we have reprocessed and reanalysed the data using BioXTAS RAW and ATSAS to provide appropriate statistical treatment of the data, as described in the methods. Error reporting is included throughout tables and figures.

5) It is necessary to quote the errors on R_g in both the text and Tables (e.g., Table 1, Supp Table 2, etc)!

The reanalysis of data has generated the appropriate error estimates for R_g , as well as assessing the fits of the $P(r)$ distributions, which were included in the Tables.

6) It is stated that:
“Error-weighted residuals (Δ/σ) were calculated using $I(\text{exp})-I(\text{model})/\sigma$, and are displayed as lines after fast Fourier transform smoothing.”

What is fast Fourier transform smoothing and why would you need to do this if the smoothed fit to the data is already provided from EOM? You’d simply calculate the error-normalized residual plot using this smoothed fit.

We have removed FFT smoothing from the analysis.

7) For the $p(r)$ analysis, although the authors quote R_g and D_{max} values, the issue is:

- i) I cannot find any plots of the $p(r)$ comparing the constructs. A comparative plot should be included, where the $p(r)$ are all scaled for direct comparison, for example scale the data to a common point (e.g., a scaled overlay), or normalize $I(0)$ to 1 and calculate $p(r)$, etc.
- ii) I cannot find how the authors assessed the fit to the data of the calculated $p(r)$ in reciprocal space i.e., reciprocal-space fits of $p(r)$ to the data. You understand the issue here, if the $p(r)$ do not fit the

experimental data, then comparing Rg and Dmax values has little meaning! So, reciprocal space fits to the data and quality of the fits needs to be included, and reported in the Tables.

We agree and have added two additional figures to the supplementary material comparing the P(r) plots for each of the constructs (Figure S10 and Figure S14 of the resubmission). The SAXS analysis tables were updated to give information on the reciprocal space fit to the SAXS data.

8) It is stated that:

“Together, our observations confirm that our disulfide- engineered variants evoke a more restricted degree of conformational flexibility, as well as greater compactness, compared to the parental C232S kC214S variant.”

Sorry, I do not agree with this based on the data presented in the manuscript (or alternatively caused by my confusion about what construct is being referred to).

When I compare the following three SAXS profiles directly for the ChiLob7/4 variant, that have all been scaled together relative to point 20 of each data file:

SASDSC7 hIlgG2 C232S+T222C kE123C+C214S
SASDSD7 hIlgG2 C232S+ K228C kC214S
SASDUL8 hIlgG2 C232S kC214S

After scaling and comparing, these profiles are statistically identical. Do the authors not find this interesting considering all three contain hIlgG2 C232S kC214S...(although this latter mutation is also listed as C214S and not kC214S, mutation in Table S6. Is there a difference between kC214S and C214S?) In addition SASDUH8 (hIlgG2) is also very similar to the three entries listed above when you scale the data to the 20th point or so (could be incorrect background subtraction)...

If SASDSC7, SASDSD7, SASDUL8 and potentially SASDUH8 (after correct background subtraction) are indeed identical then how does this direct observation influence the statement:

“Primary analysis of the SAXS data indicated that both designed variants were slightly more compact than the parental C232S kC214S in solution, with decreased Rg and Dmax values in the distance distribution function.”

This reviewer fundamentally does not agree. Within error all three(four) profiles and the subsequent structural parameters extracted from the SAXS data – especially the Rg – are effectively the same, indeed all engineered mutant construct (hIlgG2 C232S+T222C kE123C+C214S, hIlgG2 C232S+ K228C kC214S and hIlgG2 C232S kC214S) are similar or the same as SASDUH8 (hIlgG2) within measurement error. The only differences I can see in the ChiLob7/4 cohort are:

SASDUG8 hIlgG1 – is unique
SASDUJ8 hIlgG2 C232S+ C233S – is unique
SASDUK8 hIlgG2 C233S kC214S – is unique

Then SASDSC7, SASDSD7, SASDUL8 and probably SASDUH8 can probably be grouped together as the same (effectively).

We thank the reviewer for their analysis of the data and the excellent suggestions. We performed correlation analysis, and the results support the concern of the reviewer. We removed statements on conformational flexibility and caution on the analysis in the discussion (from line 388 of the original submitted manuscript):

We demonstrate that the designed disulfides result in a rigid conformation similar to the parent molecule C232S κC214S (cross-over). While the new variants are slightly more compact than the parent, these differences were subtle and difficult to evaluate with current methodology. Despite this, we show that even small structural and conformational differences in these molecules can elicit substantial differences in agonistic activity, with the two engineered variants exhibiting significantly greater biological activity than the parent.

Note 1) One cannot resort to Dmax as an estimator of difference, unless Dmax is hugely significantly different. Please remember that Dmax is just a number to solve an integral. It is not directly measured. Consequently it has a huge error, especially if the data do not transform well at high values of r – probably +/-5 angstroms at best. This impacts e.g., the presentation of the plot in Figure 3 e.

We agree with the reviewer and have removed Figure 3e of the original submitted manuscript.

Note 2) To really assess whether the changes in Rg noted in the text are statistically different, the authors need to go back and calculate the Rg correlation through the SEC elution peak, per subtracted frame, to obtain an idea as to the variance on Rg through the SEC elution (for all of their constructs). Only then can statements be made about differences in Rg taking into account the errors on Rg in the finally processed data.

We include calculation of the Rg correlation across the SEC peak (per subtracted frame) in the new Figures S6-S7 and S9-S10 of the resubmission.

For the other cohort of SAXS data – SAP 1.3 - that needs significant expansion and presentation in the manuscript (if it has been published before, simply referring to previous publications is insufficient here. The results need to be presented, irrespective).

See response to point 1 above for why we focus on ChiLob7/4 for the structure-guided approach. We agree with the reviewer that we must fully describe the primary SAXS analysis for SAP1.3, so we present dimensionless Kratky plots in Figure 1c and include the full SAXS analysis in Table S5 and Figure S7.

Although not statistically identical, SASDUC8, SASDUB8 and SASDUE8 are very similar (again scaling from point 20), where SASDUE8 is just hlgG2. Again, how do the additional disulphide mutations significantly affect the structure of those constructs engineered with additional mutations?

From the correlation analysis it would appear that SASDUC8, SASDUB8 and SASDUE8 are not statistically identical. The mutations affect the local structure and determine which disulfides form.

Then the authors need to do a comparative analysis of SAP 1.3 variants with ChiLob7/4 – are the SAP 1.3 variants more or less ‘flexible’ than the ChiLob7/4 variants? The point of this is to demonstrate that it is the disulphides and NOT some other effect, e.g., repositioning of surface residues induced by the disulphides, that are causing the change in conformation, or structural sampling. The authors have mentioned torsional angles, etc in the manuscript, but does this hold true for both SAP 1.3 and

ChiLob7/4?

We are not attempting to compare SAP1.3 to Lob7/4 to each other. Each group of mutants for each antibody (SAP1.3 or Lob7/4) is self-contained and validated by biological analysis. However, activity and in-solution behaviour for both sets of antibodies show the same trends.

9) The MD and EOM section.

It is the opinion of this reviewer that the MD simulations need to be revisited in order to correctly interpret the conclusions drawn in the manuscript. The major issue – and aside from C232S+C233S – is that the MD simulations are **not** sampling sufficient structures anywhere near the R_g determined from the experimental SAXS data.

It is interesting the initial pool of MD simulated structures are significantly more compact in terms of frequencies relative to the experimental data. For example when I look at Figure 3 of the main text, the most frequent R_g is about 37.5 Angstroms, well below the experimental R_g of either construct (38.6 and 38.7).

So, EOM is simply being forced to select a minimized ensemble that chooses less frequent models with higher R_g to fit the data (as the results show). The question is, would it not be better to start the MD simulation on a model that starts with an R_g similar to the R_g of the experiment – generate an ensemble from this, and then assess the sampling? This ‘forcing EOM to select limited models from restricted under-sampled pools of starting structures’ invalidates the comparison of R_{flex} between the constructs.

We agree with the concerns of the reviewer and removed the EOM analysis and show that single molecular fits for the rigidified structures are appropriate (also see comments below). The analysis has led to a major update of Figure 3 and presentation of the data in the results (from line 277 of the original submitted manuscript):

The dimensionless Kratky plots suggest subtle differences between the engineered variants and the parental cross-over (Figure 3a, Supplementary Fig. S14a). Therefore, to explore whether the variation in disulfide pattern, as seen in the crystal structures (Fig. 2c, d), would lead to discernible conformational differences, we used atomistic MD simulations to generate structural states for each variant from three replicate trajectories of 2 ms. The theoretical scattering curves were calculated for each of the atomistic models within the MD-generated conformational pools, and these curves were compared to the corresponding experimental scattering data to find the best fitting single models. For the parental cross-over variant and the two engineered variants, the best fitting single models showed good agreement with the SAXS data (Figure 3b, Supplementary Table S4), with c^2 values of 1.081 for the parent, 1.311 for the cross-over + K228C variant and 1.009 for the cross-over + T222C kE123C. These data suggested that our disulfide designs yielded antibody molecules which were conformationally consistent with the parent cross-over molecule, as can be seen from the structures of the best fitting models (Figure 3c). The good agreement of individual models extracted from the MD-generated conformational pools indicates that the engineered variants, as well as the parent, are particularly rigid, as the SAXS data can be described well with a single structure. In contrast, the best fitting single models for C232S+C233S and C233S kC214S showed poorer agreement with the SAXS data, with substantial structure seen in the error-weighted residuals and thus required ensemble fitting (Supplementary Methods and Fig. S15).”

Considering that SASDSC7, SASDSD7, SASDUL8 are identical SAXS profiles (after scaling) – and SASDUH8 is pretty close to these, in my mind there is a good argument to be made that these data could be described as *single* models. Did the authors try and simply fit single models to the data? For example in the main text of Figure 3, I think am looking at a comparison of

SASDSC7 hIlgG2 C232S+T222C κE123C+C214S
 SASDSD7 hIlgG2 C232S+ K228C κC214S

...which are statistically equivalent profiles (after scaling), but the Rflex bar graph suggests differences here, where there are none because the SAXS data are the same!

As an example of single model generation – if I take the SASBDB models deposited for SASDSC7 and SASDSD7 and simply do a CRY SOL 2.8.3 fit to the data, out to an qmax of 0.4 inverse Angstroms:

```
crystal *.pdb *.dat -lm=50 -ns=256 -sm=0.2 -un=1 -cst -err
```

For SASDSD7_fit2_model4, the χ^2 of the model fit to the data is pretty good, at 1.21, while for the other entry, SASDSC7_fit2_model1 the χ^2 is 1.24, again not that bad.

Using FoXS for the model fitting routine also provides a similar result for the single model representation fit to the data, using the maximum q-value (SASDSC7_fit2_model1, $\chi^2 = 1.26$; SASDSD7_fit2_model4, $\chi^2 = 1.3$).

...so these single model representations are not that bad, and can probably be refined even further. Why not just calculate the fits to the data of the individual models generated from MD, with a focus on those with an Rg close to that of the SAXS experiment? If you can indeed develop single models that fit the data better than the EOM models, where does this put the conclusions reached about conformational-flexibility? This needs to be investigated.

We used CRY SOL to perform fits of each of the extracted models from the MD simulations to the experimental SAXS data. Only the less restricted variants C232S+C233S and C233S κC214S benefit from the ensemble optimisation analysis. For the conformationally restricted variants, namely the parental C232S κC214S (cross-over) and the two new engineered variants, the best fits arise from single model fits for MD simulation models rather than from an ensemble optimisation approach (see table below). Thus, a single model fit is most appropriate in these cases.

	Single model χ^2	Single model Rg	GAJOE χ^2	Experimental Rg
C232S+C233S	1.607	42.34	1.028	43.56
C233S κC214S	1.232	40.45	1.193	41.12
C232S κC214Sn (cross-over)	1.081	39.57	1.087	40.00
cross-over + K228C	1.311	38.93	1.306	39.45
cross-over + T222C κE123C	1.009	39.07	1.012	39.22

10) Rg and flexibility.

As you know SAXS data are ambiguous because of the spherically averaged scattering amplitudes. Therefore it is necessary to at least provide a statement to the reader that having a shorter Rg does not

necessarily correlate with a decrease in flexibility, as implied in the manuscript, or vice-versa, having a larger R_g correlates with an increase in flexibility. There are clearly instances where modular proteins can have the same molecular weight, but be less flexible, and more extended, giving a larger R_g ; and conversely, it is conceivable that something that is on average more mass-centred has multiple mass-centred conformations (smaller R_g but still flexible).

In effect, different conformations can have the same R_g , so R_g in and of itself cannot be used to assess flexibility. How ambiguous are the data (e.g., using AMBIMETER to assess this)?

We fully agree and did not intend to imply that R_g would directly correlate with flexibility; we revised presentation of the data in results with a new paragraph (from line 304 of the original submitted manuscript):

Whilst the two engineered variants appear structurally similar to the parent cross-over, there are subtle differences in conformation, observed in the hinge angles (calculated according to Supplementary Fig. S16a), R_g and D_{max} of the best fitting single models (Fig. 3d-f, Supplementary Table S4). The designed variants are slightly more compact with hinge angles of 125.3° for the cross-over + K228C and 124.9° for the cross-over + T222C kE123C, compared to a hinge angle of 126.6° for the parent. Similarly the R_g for the best fitting model for each of the designed variants is smaller (38.93 \AA for the cross-over + K228C and 39.07 \AA for the cross-over + T222C kE123C, compared to 39.57 \AA for the parent). Together, our observations support that our engineered disulfides result in rigid and compact antibody molecules, similar to the parental cross-over variant.

Final comments:

i) If the activity binding assays etc were done on full mAb variants, the authors should also perform the same types of assays using the truncated variants used for SAXS? The Fc does sometimes affect the conformational sampling of the Fab?

We show the same trends in activity for both, $F(ab')_2$ as well as full IgG. We show the activity trends shown using full IgG (Figure 4) are equivalent to when $F(ab')_2$ fragments are used, adding a new supplementary figure (Figure S19). We have added the following text (line 359 of the original submitted manuscript):

The same trends in agonistic activity were seen using $F(ab')_2$ fragments, rather than full IgG molecules, indicating that the agonistic activity seen is Fc-independent (Figure S19).

The principle that the activity of hIgG2 antibodies is Fc-independent was demonstrated in several previous publications¹⁻⁴; the Fc is not required for activity. Specifically in our recent publication¹, we presented NF- κ B GFP reporter data as well as primary human B cell data using $F(ab')_2$ fragments as well as full IgG, and we show the same trends in activity for both formats.

ii) Maybe short statement to include that the production of the mAb do not incorporate glycans?

The antibodies are produced in mammalian CHO cells, therefore mAb production results in Fc glycosylation. This occurs in all antibodies (including natural antibodies produced in vivo) at the N297 residue in the Fc domain (unless engineered out). This glycosylation was not removed prior to our

biological assays or new IgG SAXS data collection. Lob7/4 has no glycosylation sites in its F(ab') regions so the Lob7/4 F(ab')₂ fragments assessed by SAXS do not have glycosylation, whereas SAP1.3 does have a glycosylation site in its F(ab') region. Therefore, it would be incorrect to state in the text that the production of mAb do not incorporate glycans.

iii) To me there is far too much speculation in the discussion, specifically statements that have no experimental support provided in the main text of the manuscript. For example:

“The mechanism by which conformational restriction impacts on biological activity is likely mediated by the ability of the different mAb variants to promote receptor clustering... Disulfide-engineered variants with a more compact molecular arrangement and restricted conformational flexibility may bind receptors in closer proximity, promoting efficient clustering of receptors in the cell membrane and triggering activation of downstream intracellular signalling pathways, leading to cellular activation. In contrast, mAb variants with greater conformational freedom, which sample very extended conformations, may engage two distant receptors. This would be less likely to stabilise receptor clusters and thus be incapable of surpassing the receptor signalling threshold to promote strong agonism.”

We apologise for the lack of detail with regards to where our experimental support arises from. Over the last 9 years we have assessed this family of costimulatory TNFRs and in a series of papers over that time²⁻⁶ have shown that receptor clustering is central to their mode of action. In the discussion we are simply trying to propose a plausible mechanism for how a more compact disulfide engineered antibody might achieve greater agonism. In our opinion not providing some discussion on this point would seem unusual. However, we are content with the decision of the editor if they deem this discussion too speculative.

It should also be noted that, in direct contrast, reviewer 4 requested to expand on the potential mechanism. Replying to these conflicting suggestions from the two reviewers, we have modified the discussion (from line 394 of the original submitted manuscript):

Future design of antibodies must consider the mechanism by which agonism is achieved. Conformational restriction likely impacts on biological activity by modulating receptor clustering. Agonistic activity is directly associated with receptor clustering for both anti-*h*CD40 and anti-*h*4-1BB mAb²³. The restricted conformational diversity of disulfide-engineered variants may retain receptors in closer proximity, promoting efficient clustering of receptors in the cell membrane, trigger activation of downstream intracellular signalling pathways, thus leading to cellular activation. In contrast, mAb variants with greater conformational freedom would be less likely to stabilise receptor clusters and thus be incapable of surpassing the receptor signalling threshold to promote strong agonism.

In summary, I think to be more convincing requires a reworking of the manuscript in terms of organisation and flow, along with detailed analyses of *all* of the protein variants used, which included a comparative analysis of the SAXS results.

We believe we have addressed the concerns of the reviewer by reprocessing and analysing all of the SAXS data using a different software, permitting the reporting of errors. We generated new supplemental figures detailing all the SAXS analysis for each of the constructs (i.e. SEC traces, I(q) vs q plots with Guinier fits, dimensionless Kratky plots and P(r) distributions). We have also altered the

analysis of the MD simulations in combination with the experimental SAXS data, presenting single model fits. We have refined our reference to flexibility and conformational restriction.

Changes or additions were made to Figure 1c, Table 1, Figure 3, Table S2, Table S4, Table S6, as we added or updated Figures S6-S10, S14-15 and updated or added Tables S4-S7 and S11-S12. We have edited the text in accordance with these changes. We hope the additions and changes have improved the understanding and flow of the text.

Reviewer #4 (Remarks to the Author):

In this manuscript, the authors show how disulphide engineering can enhance the agonist activity of antibodies. The hypothesis is that the effect is due to restricted flexibility in the antibody. The authors probe the flexibility using SAXS and atomistic Molecular Dynamics simulations. Due to the rather small changes in the size of the antibodies, they work with Fab₂ fragments.

This aspect of the manuscript raises significant questions that the authors will need to satisfactorily address:

To clarify the validity of the approach: we have previously shown that F(ab')₂ fragments are as active as IgG for hIgG2 ChiLob7/4 (and several other antibodies in the hIgG2 isotype)¹⁻⁴. Given there are no differences in activity, we utilise the reduced complexity of the F(ab')₂ for SAXS and MD analysis.

1. Is the behaviour of the solvated fragments representative of that in the full antibody, where the Fc will sterically influence the relative Fab positions and their dynamics? This aspect needs more discussion.

We have performed the activation assays in Figure 4d-f with full hIgG and have now also performed these with F(ab')₂ fragments (Figure S19 in the resubmission). We show that the trends in primary B cell activation are the same in either format. This is in line with the previous observation¹⁻⁴.

Conversely, we also performed SAXS analysis for F(ab')₂ fragments and compare this with data obtained for full hIgG. The differences are smaller in the IgG format (Table S6 and S7) but follow the same trends as seen for F(ab')₂ fragments (Table S4 and S5), in particular for the P(r) distributions. Similarly, the trends are visible from comparison of the Kratky plots (Figures S6-S7 and S9-S10).

2. In the simulation work, the authors report “reweighted ensembles” without proper explanation of what this means and why it is justified. On the face of it, it seems like they select simulated conformations that align with experimental data, and discard the rest, so should we be surprised when the end results agree?

This analysis has been removed. See comments to other reviewers.

3. The changes to the size of the antibody fragments are small, yet data is presented without statistical analysis (not even errorbars in Table 1). This must be properly addressed.

All SAXS data have been re-processed and now include statistical analyses and error estimates. See also detailed response to reviewer 3.

4. The hypothesis that the more rigid constructs promote receptor clustering could be complemented by simulation data for the relative CDR conformations. It seems unlikely to me that reduced flexibility would enhance receptor interaction, however some further insight might be possible to support the picture.

Modelling of the CDR regions are not helpful as they are unchanged across the variants. The mutations presented affect conformation of the hinge. We appreciate that receptor clustering may be an attractive hypothesis to explain increased activity, and have revised the earlier discussion:

Future design of antibodies must consider the mechanism by which agonism is achieved. Conformational restriction likely impacts on biological activity by modulating receptor clustering. Agonistic activity is directly associated with receptor clustering for both anti-hCD40 and anti-h4-1BB mAb23. The restricted conformational diversity of disulfide-engineered variants may retain receptors in closer proximity, promoting efficient clustering of receptors in the cell membrane, trigger activation of downstream intracellular signalling pathways, thus leading to cellular activation. In contrast, mAb variants with greater conformational freedom would be less likely to stabilise receptor clusters and thus be incapable of surpassing the receptor signalling threshold to promote strong agonism.

Analysis of receptor clustering in relation to conformational restriction is technically demanding and not an experiment that can simply be “added”. Referee 3 commented under point (iii) that the original discussion was too extensive and speculative. While we feel that clustering must be discussed, on balance we decided to shorten this section.

Consequently, I do not recommend accepting the manuscript in its current form.

We hope that we have addressed many of the reviewers’ concerns with inclusion of new data and with significant modification of the manuscript, as seen from the point-by-point response to all reviewers.

References

- 1 Orr, C. M. *et al.* Hinge disulfides in human IgG2 CD40 antibodies modulate receptor signaling by regulation of conformation and flexibility. *Science Immunology* **7**, eabm3723, doi:10.1126/sciimmunol.abm3723 (2022).
- 2 Yu, X. *et al.* Complex Interplay between Epitope Specificity and Isotype Dictates the Biological Activity of Anti-human CD40 Antibodies. *Cancer Cell* **33**, 664-675, doi:10.1016/j.ccell.2018.02.009 (2018).
- 3 Yu, X. *et al.* Isotype Switching Converts Anti-CD40 Antagonism to Agonism to Elicit Potent Antitumor Activity. *Cancer Cell* **37**, 850-866, doi:10.1016/j.ccell.2020.04.013 (2020).
- 4 White, A. L. *et al.* Conformation of the Human Immunoglobulin G2 Hinge Imparts Superagonistic Properties to Immunostimulatory Anticancer Antibodies. *Cancer Cell* **27**, 138-148, doi:10.1016/j.ccell.2014.11.001 (2015).
- 5 Yu, X. *et al.* TNF receptor agonists induce distinct receptor clusters to mediate differential agonistic activity. *Communications Biology* **4**, 772, doi:10.1038/s42003-021-02309-5 (2021).
- 6 Yu, X. *et al.* Reducing affinity as a strategy to boost immunomodulatory antibody agonism. *Nature* **614**, 539-547, doi:10.1038/s41586-022-05673-2 (2023).

Point by Point response to the Reviewers' Comments:

Reviewer #1 (Remarks to the Author):

All my comments have been addressed and I endorse publication of this important work.

Thank you.

Reviewer #2 (Remarks to the Author):

In the revised version of the manuscript, the authors have added the data on their novel antibody constructs demonstrating their monomeric behavior in SEC-HPLC in native conditions, and their high thermal stability measured with DSF, both of which supports the high developability potential of their proposed format and is relevant to increase the importance of this study with immanent high translational value. Further, they have improved on concept presentation (Figure 1a), and reworded the findings for more semantical clarity. The study is an important contribution to the knowledge base on engineering of antibodies for modulation of biological effects (in their case agonism) by altering the mobility of the arms rather than altering the binding sequence, which is of interest for several targets with validated clinical relevance (e.g. most in immunoncology).

Will the work be of significance to the field and related fields?

Yes

Does the work support the conclusions and claims, or is additional evidence needed?

Yes, well supported

Are there any flaws in the data analysis, interpretation and conclusions? Do these prohibit publication or require revision?

No

Is the methodology sound? Does the work meet the expected standards in your field?

Yes

Is there enough detail provided in the methods for the work to be reproduced?

Yes

Minor: line 92, something with the reference inline citations

Thank you, updated.

Reviewer #3 (Remarks to the Author):

....and that's how you do it.

I do not know who is responsible for reprocessing and analysing the SAXS data, modelling, interpretation and presentation, but they should be commended. Basically a transformative text when compared to the first version of the manuscript, so well done.

IGE, HF, IT, JWE and IT did the update. Thank you for the comment.

i) I only have very minor comments, all are in relation to just double checking consistency across text, tables and for the SAXS the parameters reported in SASBDB. Just go through all of the numbers and parameters (a tricky task for sure), e.g., the Rg quoted in the tables, text, figures, Figure panels and SASBDB etc, to make sure all are consistent for each construct.

Thank you. All figures have been checked, and SASBDB records have been updated.

ii) Tiny things like, in the Methods: 'SAXS structure and fits' it is quoted:

"...experimental SAXS data (cut to $q = 0.2$)"
--the unit on q is missing.

Thank you, updated.

Another example: In the Supp Info, Figure S12, at the end of the legend it refers the reader to Figure S12.

Thank you, reference updated to ref to Figure S13.

All of this can be done with careful re-reading, and input from the Senior editor.

iii) Only one point I would like to see in the text: A short statement somewhere about glycosylation. I brought up the topic of glycans in my review because I have a major bug about glycosylation in general. People often omit the glycosylation state(s) in their SAXS models, because they think that glycans miraculously do not scatter X-rays (which is of course complete nonsense). Of course, and especially for anything antibody-related, glycosylation could affect both stability and conformational sampling. Therefore, it would be good to include a statement somewhere in the text along the lines of how you responded to me, e.g.,:

"The antibodies are produced in mammalian CHO cells, therefore mAb production results in Fc glycosylation. This occurs in all antibodies (including natural antibodies produced in vivo) at the N297 residue in the Fc domain. This glycosylation was not removed prior to our 14 biological assays or new IgG SAXS data collection. Lob7/4 has no glycosylation sites in its F(ab') regions so the Lob7/4 F(ab')₂ fragments assessed by SAXS do not have glycosylation, whereas SAP1.3 does have a glycosylation site in its F(ab') region."

We fully agree this is important. We have revised the methods section on Antibody production and purification and added the following paragraph:

Antibodies used in this study were produced in mammalian CHO cells, resulting in Fc glycosylation. This occurs in all antibodies (including natural antibodies produced *in vivo*) at the N297 residue in the Fc domain. Glycosylation was not removed prior to our biological assays or IgG SAXS data collection. Glycosylation can also occur in the F(ab') regions. ChiLob7/4 has no glycosylation sites in its F(ab') regions, so the ChiLob7/4 F(ab')₂ fragments do not include glycosylation, whereas SAP1.3 does have a glycosylation site in its F(ab') region so the SAP1.3 F(ab')₂ fragments may include glycosylation.

iv) Okay, I let you get away with the statement:

The good agreement of individual models extracted from the MD-generated conformational pools indicates that the engineered variants, as well as the parent, are particularly rigid, as the SAXS data can be described well with a single structure.

---not technically true in and of itself (ask your SAXS expert why). You might want to re-word?

Thank you; we have taken advice from our SAXS expert (HF) and reworded this section:

The good agreement of individual models extracted from the MD-generated conformational pools indicates that the engineered variants, as well as the parent, predominantly adopt a single conformation or a highly restricted conformational ensemble under the conditions tested, allowing the SAXS data to be described well by a single representative structure.

I still think that - and although greatly de-emphasised relative to the first version of the manuscript - the statement in the discussion:

"The restricted conformational diversity of disulfide-engineered variants may retain receptors in closer proximity, promoting efficient clustering of receptors in the cell membrane, trigger activation of downstream intracellular signalling pathways, thus leading to cellular activation. In contrast, mAb variants with greater conformational freedom would be less likely to stabilise receptor clusters and thus be incapable of surpassing the receptor signalling threshold to promote strong agonism."

...is a bit dangerous to conclude based on the data presented *this* study....You really are looking at having to do some sort of reconstituted liposome/receptor + fluorescence microscopy work here to back up this statement. Receptor clustering is assumed, but there is no data in the manuscript to support the clustering hypothesis. I of course do not exclude this hypothesis, but it would not shock me if another process within or at the surface of the cell could be involved.

We appreciate the reviewer's caution but would observe that in our studies of antibody mediated receptor agonism over the last decade activity is directly correlated with receptor cross-linking, currently with no exceptions (e.g. White JI 2014, Yu Can Cell 2018, 2020, Yu et al Comms Biol 2022, Heckels, Comms Biol 2023, Yu et al Nature 2023). This is in agreement with the physiological and widely accepted mechanism of action through trimeric ligands cross-linking monomeric receptors.

And then there is the stability of the antibodies to consider, not in terms of thermodynamic stability, but something like resistance to secondary proteolysis/processing events, redox-state(s), blah blah blah.

Again, we appreciate the (somewhat hypothetical) concerns presented but would note that the kinetics of cross-linking are rapid – occurring as rapidly as within 10 minutes, reducing the likelihood of their impact. Moreover, as far as we are aware, no evidence has been presented to date to suggest these activities are involved.

It all depends on how the antibodies engage with the receptors correct? Where they bind and in what stoichiometry(s) they bind, what effect they have on receptor conformation, and maybe even recruitment of secondary/tertiary components in/along the cell membrane (that might not be the actual targeted receptor) - could be a combination of all of this together? Basically it is complicated. I leave this over to the editor to decide.

We do agree that antibody mediated receptor cross-linking is complex. Although many other processes may impact this phenomenon we wanted to provide the most likely hypothesis. In this, we have applied Occam's razor to provide the simplest model that would explain the experimental data. We feel that cell and receptor biologists would value this hypothesis as something amenable to testing.

In conclusion - check consistency, tidy up small stuff here and there. Should be fine.

Thank you very much for your detailed comments, which we agree has massively improved the presentation and interpretation of the SAXS data, and thus our manuscript.

Reviewer #4 (Remarks to the Author):

The authors have responded positively to my objections. I have no further qualms about proceeding to publication with this paper, other to note that the authors still state some results without corresponding +/- SD (e.g. page 7 hinge angles, Rg values).

Thank you. During the first revision the presentation was updated and Table S4 now shows single model fits, hence hinge angles and predicted Rg values are given as values w/o SD.